# A small periplasmic protein governs broad physiological adaptations in *Vibrio cholerae* via regulation of the DbfRS two-component system

Emmy Nguyen [1], Charles Agbavor [2], Anjali Steenhaut[3], M. R. Pratyush [1], N. Luisa Hiller [1], Laty A. Cahoon [2], Irina V. Mikheyeva[1], Wai-Leung Ng[3] & Andrew A. Bridges [1] ✉

Two-component signaling pathways allow bacteria to sense and respond to environmental changes, yet the sensory mechanisms of many remain poorly understood. In the pathogen *Vibrio cholerae*, the DbfRS two-component system controls the biofilm lifecycle, a critical process for environmental persistence and host colonization. Here, we identified DbfQ, a small periplasmic protein encoded adjacent to *dbfRS*, as a direct modulator of pathway activity. DbfQ directly binds the sensory domain of the histidine kinase DbfS, shifting it toward phosphatase activity and promoting biofilm dispersal. In contrast, outer membrane perturbations, caused by mutations in lipopolysaccharide biosynthesis genes or membrane-damaging antimicrobials, activate phosphorylation of the response regulator DbfR. Transcriptomic analyses reveal that DbfR phosphorylation leads to broad transcriptional changes spanning genes involved in biofilm formation, central metabolism, and cellular stress responses. Constitutive DbfR phosphorylation imposes severe fitness costs in an infection model, highlighting this pathway as a potential target for anti-infective therapeutics. We find that *dbfQRS*-like genetic modules are widely present across bacterial phyla, underscoring their broad relevance in bacterial physiology. Collectively, these findings establish DbfQ as a new class of periplasmic regulator that influences two-component signaling and bacterial adaptation.

Microorganisms must adapt to environmental fluctuations to ensure their persistence. As a result, over evolution, bacteria have developed sophisticated sensory mechanisms that translate environmental signals into adaptive responses. Among these mechanisms, two-component systems (TCSs), which consist of a membrane-associated sensor histidine kinase and a cognate response regulator, are ubiquitous across the bacterial domain[1]. Upon the detection of specific stimuli, histidine kinases modify the phosphorylation state of the response regulator, which, in turn, orchestrates physiological changes, often through regulation of gene expression[2,3]. The prevalence and diversity of TCSs in bacterial genomes is immense, with some individual species encoding as many as 200 distinct TCSs[4]. Despite their pervasiveness, research into TCS molecular mechanisms has primarily concentrated on a handful of major TCS families[5–7]. As a result, for

[1]Department of Biological Sciences, Carnegie Mellon University, Pittsburgh, PA, USA. [2]Department of Biological Sciences, University of Pittsburgh, Pittsburgh, PA, USA. [3]Department of Molecular Biology and Microbiology, Tufts University School of Medicine, Boston, MA, USA. ✉e-mail: bridges@cmu.edu

most TCSs, the identities of their stimuli, mechanisms of signal transduction, and consequences to bacterial physiology remain underexplored.

TCSs frequently control bacterial social behaviors, including regulation of the biofilm lifecycle, whereby bacteria form surface-associated, multicellular communities[8–10]. In the biofilm state, bacterial cells are encased in a self-produced extracellular matrix that confers environmental protection against antimicrobials, immune responses, fluid flow, and bacteriophage predation[11–13]. For these reasons, biofilms are notoriously difficult to eradicate in both clinical and industrial settings. Despite the adaptive benefits of the biofilm lifestyle, prolonged biofilm formation can be associated with bacterial fitness costs. Dense multicellular communities face competition for resources, their surface-associated characteristic limits spread to new territories, and extracellular matrix production diverts resources from metabolic processes[14,15]. Consequently, bacteria have evolved intricate sensory mechanisms, often involving TCSs, to control the balance of biofilm formation and biofilm dispersal, whereby cells transition to the free-swimming state[16–18].

For *Vibrio cholerae*, the causative agent of cholera disease, the biofilm lifecycle is thought to underlie its ability to transition between the aquatic niche and the human host[19,20]. Moreover, transitions between bacterial niches are often mediated by TCSs, enabling host adaptation in facultative pathogens such as *V. cholerae*[21,22]. In prior work, we identified a TCS, named DbfRS (encoded by *vc_1638* and *vc_1639*), as a regulator of *V. cholerae* biofilm lifecycle transitions[23]. Within this cascade, DbfS serves as the sensor histidine kinase that controls the phosphorylation state of the transcription factor DbfR (Fig. 1a). The balance between kinase and phosphatase activities of DbfS determines the phosphorylation state of DbfR, which in turn dictates whether cells commit to the biofilm state or disperse. Specifically, dephosphorylation of DbfR enables biofilm dispersal, whereas DbfR phosphorylation activates biofilm formation and represses dispersal via positive regulation of biofilm matrix gene expression (Fig. 1a). Beyond these basic findings, the DbfRS pathway remains uncharacterized—the environmental signal(s) controlling pathway activity remain unknown, the DbfR regulon is undefined, and its role during infection is unclear.

In this study, we comprehensively investigated the DbfRS regulatory network, defining its sensory inputs, physiological outputs, and fitness consequences. We identified a previously uncharacterized small protein, DbfQ, that directly interacts with and modulates the sensory domain of DbfS, driving the receptor toward phosphatase activity. We find that outer membrane stresses activate the DbfQRS pathway and that DbfR phosphorylation leads to global transcriptional changes, coordinating biofilm commitment with reduced metabolic activity. Activation of the DbfQRS pathway is detrimental to bacterial colonization in an animal model of infection, highlighting the potential of this pathway as a therapeutic target for limiting the spread of *V. cholerae*. Finally, the widespread occurrence of dbfQRS-like genetic modules, including in other important pathogens, underscores the broader relevance of this novel regulatory mechanism.

## Results

### A small protein controls the biofilm lifecycle via regulation of the DbfRS signaling cascade

For many bacterial TCSs, the genes encoding response regulators and histidine kinase pairs commonly exist within the same operon. Examination of the *dbfRS* operon revealed a third gene, herein referred to as *dbfQ*, encoded immediately upstream and predicted to be co-transcribed with *dbfRS* (Fig. 1b)[24]. Given its presumed co-regulation, we wondered whether DbfQ could participate in regulating the biofilm lifecycle via the DbfRS signaling pathway. *dbfQ* encodes a 135-amino-acid protein with a predicted N-terminal secretion signal (Sec/SPI)[25]. The mature ~12 kDa protein is predicted to contain a single domain,

identified as "PepSY" (residues 74–122) in the Pfam database[26]. Several studies have implicated PepSY domains in metal binding and/or protease inhibition[27,28], though the exact function of this motif is unclear. To determine whether DbfQ modulates the biofilm lifecycle, we constructed an in-frame deletion of dbfQ. Similar to the ΔdbfS mutant, the ΔdbfQ strain exhibited enhanced biofilm formation and a defect in biofilm dispersal, with substantial biofilm biomass remaining at the final time point (Fig. 1c). Both mutants also exhibited a growth defect compared to wild-type (WT) (Supplementary Fig. 1a). Complementation of the ΔdbfQ mutant by introduction of P$_{tac}$-dbfQ at a neutral locus restored the biofilm lifecycle and growth rate to that of WT (Supplementary Fig. 1b–e). Our next goal was to determine whether DbfQ controls the biofilm lifecycle through the DbfRS signaling cascade. To assess this possibility, we introduced the ΔdbfQ deletion into a genetic background carrying an allele of DbfR that is incapable of phosphorylation (the phospho-dead allele dbfR$^{D51V}$). We reasoned that if the extreme biofilm phenotype of ΔdbfQ is due to elevated phosphorylation of DbfR, then combining this mutant with the phospho-dead allele should abolish this phenotype. Indeed, we found that the ΔdbfQ dbfR$^{D51V}$ double mutant lost the hyper-biofilm phenotype observed in ΔdbfQ single mutant (Fig. 1d), supporting a model in which DbfQ regulates biofilm dynamics through the DbfRS signaling pathway, potentially by modulating DbfS activity.

Bacterial TCSs show a broad range of conservation, with some TCSs exhibiting a high degree of similarity across diverse taxa, whereas others are present only in specific lineages[29]. Given this understanding, we wondered whether DbfQ-like proteins encoded adjacent to TCSs are widespread. To investigate the pervasiveness of the DbfQRS module, we conducted a bioinformatic search for gene neighborhoods in which a small PepSY domain-containing protein was encoded in the vicinity of a histidine kinase or response regulator (vicinity was defined as within four genes). We found that such gene neighborhoods were present across multiple bacterial phyla, with a large representation of Proteobacteria as well as instances in Terrabacteria and Fusobacteria (Fig. 1e and Supplementary Fig. 2a). Indeed, the observed frequency of a PepSY-domain protein encoded in proximity to a TCS was >60-fold higher than expected by chance (see "Methods"). Notably, DbfQRS-like modules are present in numerous bacteria associated with human infections and agriculture. The organization of these genetic modules varied, with, in some cases, two DbfQ-like proteins encoded upstream of the TCS (e.g., *Pseudomonas aeruginosa*) (Fig. 1e). In other cases, PepSY-protein and TCS genes were adjacent but encoded on opposite strands (e.g., *Burkholderia pseudomallei*) (Fig. 1e). Collectively, these findings show that DbfQRS-like genetic modules are widespread, suggestive of a potentially conserved regulatory principle at play across diverse bacteria.

Given the dramatic effect of DbfQ on the biofilm lifecycle of *V. cholerae*, combined with the widespread presence of DbfQRS modules, we sought to determine the molecular mechanisms by which DbfQ impinges upon the TCS. Prior work demonstrated that DbfS functions as a phosphatase under laboratory growth conditions, permitting biofilm dispersal[23]. In the ΔdbfS mutant, DbfR is phosphorylated by an unidentified kinase or small molecule phosphate donor, leading to increased biofilm formation and inhibited dispersal. We set out to confirm this relationship using Phos-tag gel analysis, where more negatively charged, phosphorylated species migrate slower than the dephosphorylated form. Consistent with earlier findings, in the absence of *dbfS*, a distinct second, slower-migrating band corresponding to phosphorylated DbfR was detected (Fig. 1f). As histidine kinases typically switch between kinase and phosphatase modes depending on the ligand occupancy of their sensory domains, we reasoned that DbfQ could influence the balance between these catalytic activities of DbfS. To test this possibility, we examined DbfR phosphorylation in the ΔdbfQ mutant strain and observed complete phosphorylation of the DbfR response regulator (Fig. 1f). In contrast, the ΔdbfQ ΔdbfS double mutant phenocopied the ΔdbfS single mutant,

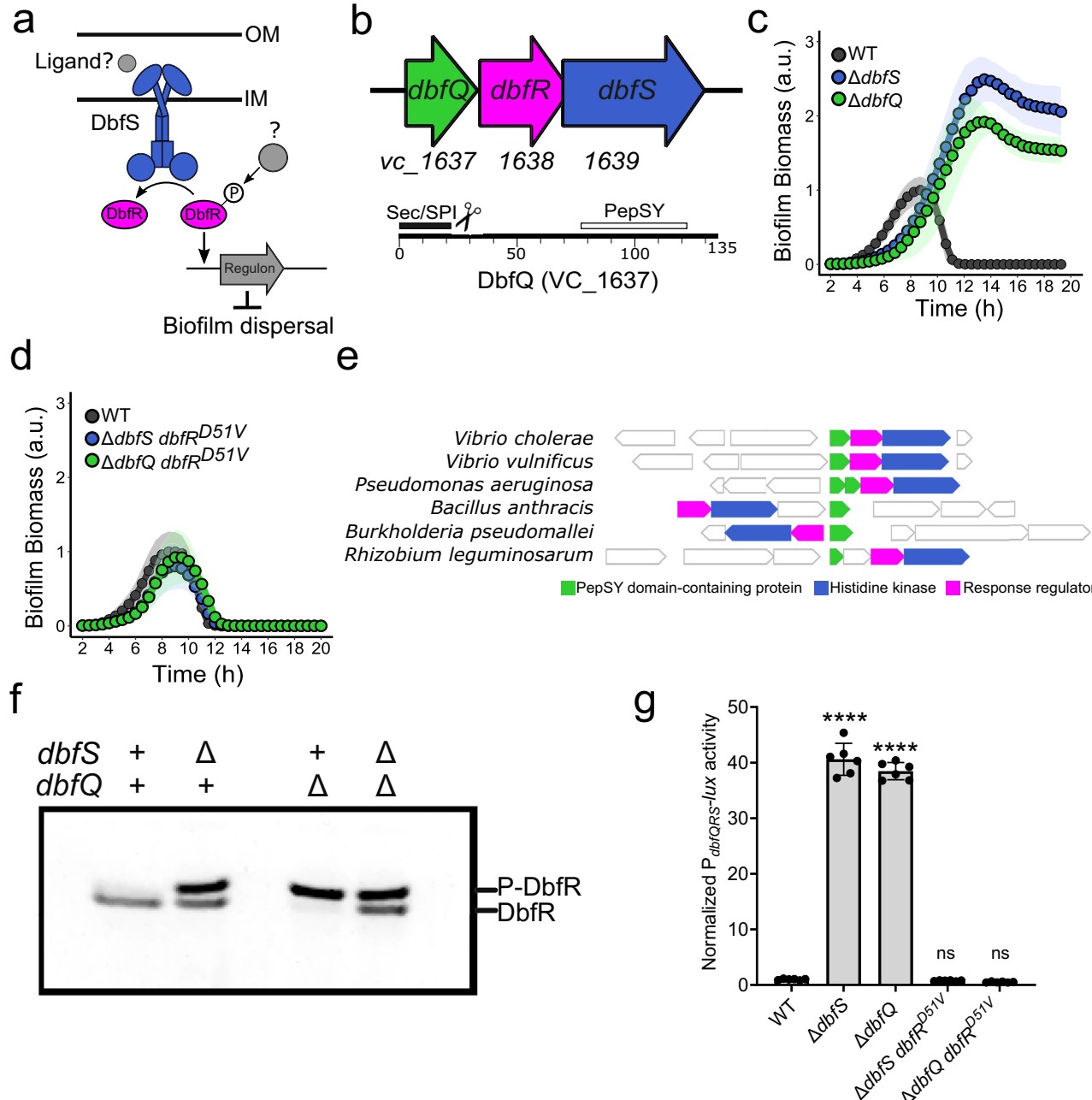

**Fig. 1 | DbfQ regulates the biofilm lifecycle via modulation of the DbfRS signaling cascade. a** Initial model for DbfRS regulation of the biofilm lifecycle. OM outer membrane, IM inner membrane. Gray objects remain uncharacterized. **b** Top panel: Operon structure of the genes encoding the DbfRS pathway. The *vc_1637* gene encoding a small hypothetical protein DbfQ is located directly upstream of *dbfR* and *dbfS*. Bottom panel: Domain architecture of DbfQ. **c** Quantification of biofilm biomass over time for WT *V. cholerae*, Δ*dbfQ*, and Δ*dbfS* mutants using time-lapse brightfield microscopy. Points represent the means ± standard deviations (SD, shaded regions) of *N* = 2 biological replicates and 3 technical replicates. **d** As in (**c**) for the Δ*dbfS dbfR^DS1V^* and Δ*dbfQ dbfR^DS1V^* strains. **e** Gene neighborhood analysis reveals a module of PepSY domain-containing protein(s) (green), histidine kinase

(blue), and response regulator (magenta) genes encoded nearby in the genome. A selected set of species associated with human infections or agriculture is shown. **f** Phos-tag analysis of DbfR-SNAP in-gel fluorescence in the presence (+) and absence (Δ) of *dbfS* and *dbfQ*. **g** Quantification of P*dbfQRS*-*lux* reporter activity for the indicated strains. Data are presented as means ± SD of peak relative light units (RLU, defined as *lux*/OD$_{600}$) normalized to the average peak value of WT. Points represent individual replicates of *N* = 2 biological replicates and three technical replicates. Statistical analysis was performed using one-way ANOVA ($P < 10^{-5}$) with Dunnett's multiple comparisons test to compare each mutant to WT. $P < 10^{-5}$, $<10^{-5}$, = 0.9960 and = 0.9453 for the indicated pairs. ****, $P < 0.0001$; ns not significant. a.u. arb. units.

demonstrating that DbfQ controls DbfR phosphorylation and the biofilm lifecycle by modulating the kinase-phosphatase equilibrium of DbfS. Because DbfR is phosphorylated in the absence of DbfQ, our results suggest that DbfQ biases DbfS towards phosphatase activity. Together, these results demonstrate that DbfQ acts as a key regulator of the DbfRS signaling cascade, impinging on the biofilm lifecycle by controlling the activity of the DbfS receptor.

To establish a quantitative readout of DbfQRS signal transduction, we set out to generate a luminescent reporter for pathway activity. During Phos-tag analysis (Fig. 1f), we noticed that increased DbfR phosphorylation appeared to correlate with elevated DbfR levels, suggestive of positive autoregulation of the *dbfQRS* operon, a common feature of bacterial TCSs[2]. Thus, to generate a luminescent reporter for DbfR phosphorylation, we fused the *dbfQRS* promoter to luciferase

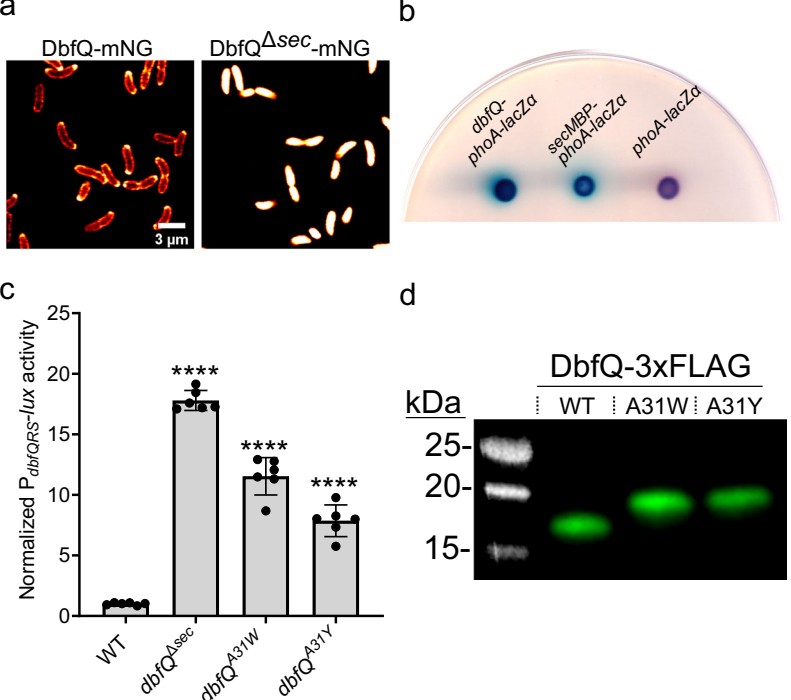

**Fig. 2 | Periplasmic secretion of DbfQ is essential for its regulation of DbfS activity. a** Left: Representative confocal microscopy images of DbfQ-mNG in *V. cholerae* expressed from an ectopic locus. Right: As in left panel, except the secretion signal of DbfQ has been removed (*dbfQ^Δsec*-mNG). **b** PhoA-LacZα fusions expressed in *E. coli* were used to assess DbfQ periplasmic localization. LacZ activity, which is non-functional in the periplasm, was assessed colorimetrically via the addition of Red-Gal (6-chloro-3-indolyl-β-D-galactoside), a beta-galactosidase substrate that turns red upon cleavage. Simultaneously, PhoA activity, which is functional in the periplasm but not the cytoplasm, is observed via the addition of X-Pho (5-bromo-4-chloro-3-indolyl-phosphate), which turns blue in the presence of phosphatase activity. Thus, redness is a measure of cytoplasmic localization, and blue pigmentation is representative of periplasmic localization. Controls included

untagged, cytoplasmically expressed *phoA-lacZα* as well as a fusion of the periplasmic secretion signal of *E. coli* maltose binding protein (MBP) to *phoA-lacZα* (*secMBP-phoA-lacZα*). Results are representative of $N = 3$ biological replicates. **c** Quantification of P*dbfQRS*-*lux* reporter activity for secretion and cleavage mutants. Data are presented as means ± SD of peak RLU normalized to the average peak value of WT. Points represent individual replicates of $N = 2$ biological replicates and 3 technical replicates. Statistical analysis was performed using one-way ANOVA ($P < 10^{-5}$) with Dunnett's multiple comparisons test to compare each mutant to WT. $P < 10^{-5}$ in all cases. ****$P < 0.0001$. **d** Western blot analysis of the FLAG-tagged WT DbfQ (cleaved) and the cleavage site mutants DbfQ^A31W and DbfQ^A31Y. Data are representative of $N = 3$ biological replicates.

(P*dbfQRS*-*lux*). To validate the reporter, we first measured luminescence in the Δ*dbfS* and Δ*dbfQ* mutants, which, as shown above, exhibit elevated DbfR phosphorylation (Fig. 1f). Both mutants displayed a ~40-fold increase in luminescence compared to the WT strain (Fig. 1g). Moreover, introduction of the phospho-dead *dbfR^D51V* allele into the Δ*dbfS* and Δ*dbfQ* mutant backgrounds reduced light production to below that of WT, demonstrating that P*dbfQRS*-*lux* reports on DbfR phosphorylation state.

**Signal peptide cleavage and periplasmic localization of DbfQ are required for its regulatory effects on DbfS**

The genetic results presented above indicate that DbfQ promotes DbfS phosphatase activity. We reasoned that DbfQ could directly interact with the DbfS receptor to modulate its activity. To investigate this possibility, we first examined DbfQ localization. As noted above, DbfQ contains a predicted N-terminal secretion signal and predicted Sec/SPI cleavage site, suggestive of secretion into the periplasm, where it could modulate the periplasmic sensory domain of DbfS. To determine DbfQ localization, we fused DbfQ to mNeonGreen (mNG) and performed confocal microscopy. While endogenously tagged DbfQ was undetectable due to low expression levels, overexpression of DbfQ-mNG revealed localization at the cell periphery, consistent with potential periplasmic localization (Fig. 2a). In contrast, an in-frame deletion of the predicted secretion signal of DbfQ (*dbfQ^Δsec*-*mNG*) resulted in DbfQ retention in the cytoplasm (Fig. 2a), suggesting that the secretion signal is essential for DbfQ localization. To confirm the periplasmic

localization of DbfQ, we fused *dbfQ* to a *phoA-lacZα* reporter, which resulted in substantial periplasmic PhoA enzyme activity (Fig. 2b)[30]. To assess whether DbfQ's periplasmic localization is required for pathway function, we examined P*dbfQRS*-*lux* output in the *dbfQ^Δsec* background and observed a >15-fold activation of light production (Fig. 2c), indicating that mislocalization of DbfQ results in increased DbfR phosphorylation. We next interrogated the role of DbfQ secretion signal cleavage by introducing bulky aromatic sidechain substitutions at the predicted cleavage residue A31 (A31Y and A31W), a strategy that, for other proteins, has been shown to inhibit processing by Signal Peptidase I[31–33]. Western blot of the WT DbfQ-3×FLAG yielded a ~17kDa band corresponding to the expected size of cleaved DbfQ-3×FLAG (Fig. 2d). In contrast, the DbfQ^A31W-3×FLAG and DbfQ^A31Y-3×FLAG exhibited higher molecular weight products consistent with the uncleaved, full-length DbfQ (Fig. 2d). The *dbfQ^A31W* and *dbfQ^A31Y* cleavage mutants exhibited 12- and 8-fold increases in P*dbfQRS*-*lux* reporter activity, respectively, compared to WT (Fig. 2c). Therefore, periplasmic localization and processing of DbfQ is required for its regulatory effect on DbfS activity, and consequently, on DbfR phosphorylation state.

**DbfQ directly interacts with the sensory domain of DbfS**

The periplasmic co-localization of DbfQ and the sensory domain of DbfS suggests that DbfQ could directly bind to DbfS to control its output activity. To explore this possibility, we first utilized AlphaFold3[34], which predicted an interaction between DbfQ and the DbfS sensory domain, with an interface predicted template modeling

(ipTM) score of 0.69, a relatively high-confidence value (Fig. 3a and Supplementary Fig. 3a). Closer inspection of the interaction interface revealed two putative electrostatic pairs, between DbfR[R84] – DbfS[D62] and DbfQ[K102] – DbfS[E69] (Fig. 3a), that could mediate the interaction. We therefore introduced site-directed mutations in either *dbfQ* (*dbfQ*[R84D] and *dbfQ*[K102E]) or *dbfS* (*dbfS*[D62R] and *dbfS*[E69K]) to weaken the interaction and test their functional relevance. Western blot analysis confirmed robust expression of the mutant proteins (Supplementary Fig. 4a), suggesting that mutagenesis did not destabilize either DbfQ or DbfS variants. We note that the observed increased levels of DbfQ and DbfS mutant proteins results from positive feedback regulation within the cascade, which amplifies loss-of-function mutant protein expression. Each point mutant yielded significantly increased P*dbfQRS*-*lux* reporter activity compared to the WT strain (Fig. 3b), suggestive of increased DbfR phosphorylation due to a weakened direct interaction between DbfQ and DbfS.

To validate the AlphaFold3 prediction of a direct interaction, we recombinantly expressed and purified DbfQ and the DbfS sensory domain (DbfS[SD]-6×His) for pull-down assays (Supplementary Fig. 4b). When supplied alone, DbfQ, which lacked the affinity tag, displayed no binding to Ni-NTA resin, whereas DbfS[SD]-6×His was bound and subsequently eluted. However, when the two proteins were pre-mixed, DbfQ co-eluted as a complex with DbfS[SD]-6×His, confirming a direct interaction (Fig. 3c). To measure binding affinity and stoichiometry, we performed microscale thermophoresis (MST). This approach revealed a high-affinity interaction between DbfQ and DbfS, with a dissociation constant ($K_d$) of 30 nM (Fig. 3d). To determine the stoichiometry of the DbfQ–DbfS interaction, the thermophoretic behavior of labeled DbfQ and unlabeled DbfS was probed using a titration to saturation experiment[35], which indicates a 1:1 stoichiometry between DbfQ and DbfS (Supplementary Fig. 4c). Collectively, these computational and experimental results establish that DbfQ controls the activity of DbfS via a direct, high-affinity interaction in the periplasm. Combined with our genetic results, we infer that DbfQ binding to DbfS biases the receptor towards phosphatase activity, driving dephosphorylation of DbfR and enabling biofilm dispersal (Fig. 3e).

Given the cross-taxa conservation of the DbfQRS system as shown by our bioinformatic analysis (Fig. 1e and Supplementary Fig. 2a), we wondered whether orthologs of DbfS and DbfQ are also predicted to interact by AlphaFold3. We focused on the system encoded in *P. aeruginosa*. Interestingly, *P. aeruginosa* possesses two DbfQ-like proteins that are encoded adjacent to a two-component system. The PepSY protein PA2658, encoded immediately upstream of the sensor BqsS, was predicted to interact with BqsS with high confidence (ipTM = 0.89, notably higher than the score for the DbfQ–DbfS prediction). In contrast, the other PepSY protein, PA2659, encoded two genes upstream, was not predicted to bind with BqsS (ipTM = 0.11), potentially suggesting a selective interaction between PA2658 and the BqsS sensory domain in *P. aeruginosa* (Supplementary Figs. 2b and 3b, c). Future studies will be required to determine whether PA2658 and PA2659 function in tandem or compete for the regulation of BqsS.

## DbfQRS activity is sensitive to changes in membrane integrity

A longstanding challenge in studying novel bacterial TCSs is identifying the signal(s) that modulate pathway activities. Some TCSs are known to respond to exogenous stimuli, such as nutrients, metals, or host-derived factors[36–38], while others integrate self-produced or intrinsic cues, including quorum-sensing autoinducers or cell envelope stress[39–41]. The *P. aeruginosa* DbfS ortholog, BqsS, was previously shown to be iron-responsive[42]; however, we found that supplementation of Fe(II) or Fe(III) did not alter P*dbfQRS*-*lux* reporter output in *V. cholerae* (Supplementary Fig. 5a). Given the tremendous number of potential exogenous inputs, we decided to focus on identifying any intrinsic factors that modulate DbfQRS activity. We reasoned that such factors could be identified using a transposon mutagenesis approach

while monitoring P*dbfQRS*-*lux* reporter activity. Since DbfS exhibits basal phosphatase activity under our laboratory conditions, we sought to identify mutations that increased P*dbfQRS*-*lux* output, reflective of elevated DbfR phosphorylation. Screening of ~20,000 Tn*5* mutagenized colonies yielded 54 isolates exhibiting elevated luminescence, which, after sequencing, mapped to 23 loci spanning eight functional categories (Table 1, Supplementary Table 1, and Supplementary Fig. 5b). Among the hits, the identification of *dbfQ*::Tn*5* and *dbfS*::Tn*5* validated our screening strategy, as these mutants display increased DbfR phosphorylation (Fig. 1f, g). Strikingly, the largest category of hits (~35%) mapped to genes predicted to be involved in lipopolysaccharide (LPS) biosynthesis. Defects in LPS biosynthesis can profoundly affect the cell envelope by increasing membrane permeability, altering membrane protein expression, and affecting susceptibility to antibiotics[43,44]. Consistent with our findings, previous work implicated *vc_1639* (*dbfS*) in resistance to polymyxin B[45], a cationic antimicrobial peptide that disrupts the LPS network and destabilizes the outer membrane. These results suggest a relationship between membrane integrity and DbfQRS pathway activity. To validate this relationship, we first constructed an in-frame deletion of *wavA* (*vc_0223*), the first gene in the LPS core-polysaccharide biosynthesis cluster, which was previously reported to result in truncated LPS lacking O-antigen attachment[46,47]. The Δ*wavA* mutant exhibited an 8-fold increase in luminescence compared to WT strain, consistent with elevated DbfR phosphorylation (Fig. 4a). Introducing the phospho-dead *dbfR*[D51V] allele into Δ*wavA* background suppressed light production to below that of WT, confirming that modifications in LPS enhance pathway activation specifically via DbfR phosphorylation, and not via parallel transcriptional control of the *dbfQRS* promoter. To further substantiate the link between membrane stress and DbfQRS activation, we treated WT cells with sub-lethal concentrations of the membrane-targeting antibiotic polymyxin B and measured P*dbfQRS*-*lux* reporter output. This treatment led to a dose-dependent increase in luminescence, reaching a maximum of 6-fold induction at the highest concentration tested (Fig. 4b). Similarly, exposure to thymol, an antimicrobial compound known to disrupt membrane integrity[48], produced up to an 8-fold increase in luminescence (Fig. 4c). In contrast, the *dbfR*[D51V] strain failed to induce light production under either treatment, demonstrating that DbfR phosphorylation is a direct readout of outer membrane stress. Overall, our data suggest that the DbfQRS activity is responsive to perturbations in membrane integrity. Defining the precise mechanism by which DbfQRS senses membrane stress will be the topic of future work.

## The DbfQRS pathway activates biofilm formation while down-regulating metabolic processes

Our next goal was to characterize the effects of DbfQRS signaling on downstream gene expression. To define the DbfR regulon, we performed RNA sequencing under conditions where DbfR was either constitutively dephosphorylated or phosphorylated. We began by comparing the transcriptome of WT *V. cholerae* to the phospho-dead *dbfR*[D51V] strain. This strain displayed modest transcriptional changes, with 54 genes exhibiting differential expression ($Log_2FC > \pm 1.0$ and *P*-value < 0.05) (Supplementary Fig. 6a and Supplementary Data 1). In contrast, when DbfR was phosphorylated (using the Δ*dbfS* mutant strain), we observed a dramatic shift in gene expression, with 12% of genomic loci (*N* = 540 genes) exhibiting differential expression (Fig. 5a and Supplementary Data 2). These results confirm that DbfR phosphorylation drives large-scale transcriptional reprogramming. As expected, the Δ*dbfQ* mutant, which also exhibits elevated DbfR phosphorylation (Fig. 1f, g), displayed a similar transcriptional profile to the Δ*dbfS* strain (*r* = 0.93) (Fig. 5b, Supplementary Fig. 6b, and Supplementary Data 3).

To gain functional insights into the genes regulated by phospho-DbfR, we performed a KEGG pathway enrichment analysis. This

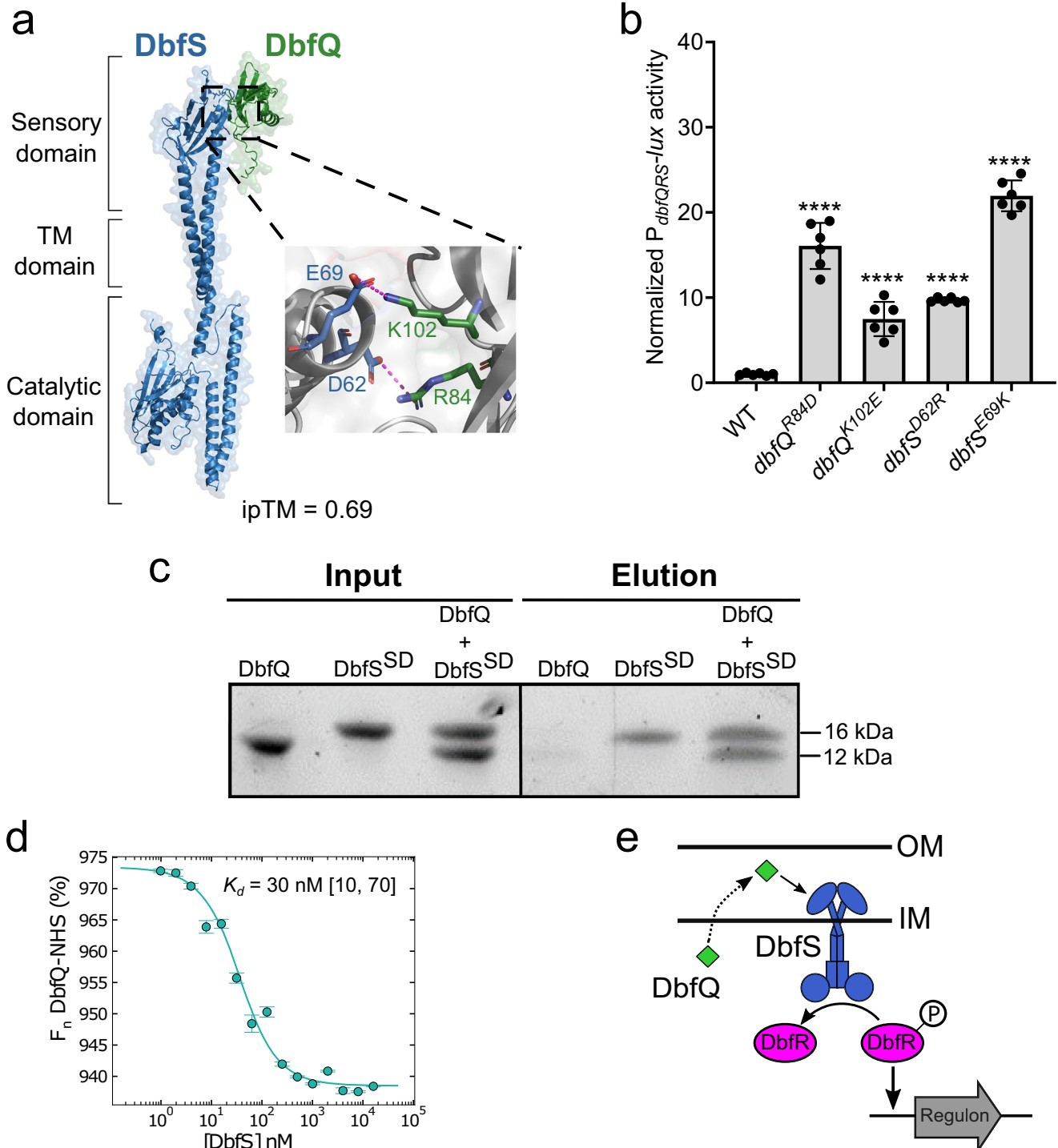

**Fig. 3 | DbfQ directly interacts with the sensory domain of DbfS. a** Left Panel: Predicted interaction between AlphaFold3 structures of cleaved DbfQ (beginning at residue 32, green) and full-length DbfS (blue), with an ipTM = 0.69. Critical domains of DbfS, including sensory, transmembrane (TM), and catalytic domains, are labeled. Right panel: Zoomed view of the electrostatic interactions (magenta dashed line) formed between DbfQ$^{R84}$ – DbfS$^{D62}$ and DbfQ$^{K102}$ – DbfS$^{E69}$. **b** Quantification of P$_{dbfQRS}$-*lux* reporter activity for the indicated *dbfQ* and *dbfS* point mutants. Data are presented as means ± SD of peak RLU normalized to the average peak value of WT. Points represent individual replicates of $N = 2$ biological replicates and 3 technical replicates. Statistical analysis was performed using one-way ANOVA ($P < 10^{-5}$) with Dunnett's multiple comparisons test to compare each

point mutant to WT. $P < 10^{-5}$ in all cases. ****, $P < 0.0001$. **c** SDS-PAGE results of a Ni-NTA pull-down assay between purified DbfQ and the DbfS$^{SD}$-6×His. Data are representative of $N = 3$ independent experiments. **d** MST dose-response curve used to measure the DbfQ and DbfS$^{SD}$ interaction. DbfQ was covalently labeled with an amine-reactive Red-NHS dye and subsequently titrated against increasing concentrations of unlabeled DbfS while changes in thermophoretic behavior were monitored. Normalized fluorescence (F$_n$) results were fit with a one-site binding model (95% confidence interval), yielding a dissociation constant of 30 nM [10, 70] [Lower Limit, Upper Limit]. Data are presented as means ± SD of $N = 4$ independent replicates. **e** Proposed model for the DbfQRS signaling pathway. OM outer membrane, IM inner membrane.

**Table 1 | Functional distribution of transposon mutagenized genes with increased DbfQRS activity**

| Primary function* | No. of genes | % of genes |
|---|---|---|
| LPS biosynthesis | 8 | 35 |
| Transcription, Translation, and DNA repair | 6 | 26 |
| Transport | 2 | 9 |
| Signal transduction (dbfQ and dbfS) | 2 | 9 |
| Metabolism | 1 | 4 |
| Hypothetical proteins | 2 | 9 |
| Phage shock protein | 1 | 4 |
| Virulence | 1 | 4 |
| Total | 23 | 100 |

*Assigned by KEGG Orthology Database.

analysis revealed significant downregulation of genes involved in central metabolic pathways, including the tricarboxylic acid (TCA) cycle and oxidative phosphorylation, as well as genes implicated in purine/pyrimidine metabolism, amino acid metabolism, and the biosynthesis of secondary metabolites (Fig. 5c and Supplementary Data 4). These results are consistent with our finding that the ΔdbfS and ΔdbfQ mutants exhibited a reduced growth rate compared to WT (Supplementary Fig. 1a). Beyond metabolic changes, genes involved in flagellar assembly were also downregulated under phospho-DbfR conditions, suggesting that the pathway controls motility in addition to biofilm formation (Fig. 5a). Soft-agar motility assays confirmed that both ΔdbfS and ΔdbfQ strains exhibited significantly decreased motility compared to WT, though not to the extent of the ΔflaA mutant lacking the major flagellin subunit (Fig. 5d and Supplementary Fig. 7a). Furthermore, single-cell imaging of motility in the ΔdbfS and ΔdbfQ mutant strains revealed that a high-frequency of cells did not exhibit motility (Supplementary Fig. 7b). On the other hand, significantly upregulated pathways in the ΔdbfS and ΔdbfQ transcriptomes included genes encoding vibrio exopolysaccharide (vps) biofilm matrix components, peptidoglycan biosynthesis, and mismatch repair proteins (Fig. 5a, c). Collectively, these results suggest that DbfR activation biases V. cholerae towards a sessile biofilm state, while simultaneously slowing growth by reducing metabolic processes. We propose that entering this state could prepare cells for environmental challenges.

## DbfR phosphorylation drives biofilm formation by increasing c-di-GMP levels

The simultaneous upregulation of vps gene expression and down-regulation of motility factors observed in our transcriptomic analysis could be manifested by changes in cyclic dimeric GMP (c-di-GMP) levels. C-di-GMP is a widespread bacterial second messenger molecule which controls motile-to-sessile transitions by repressing motility and promoting biofilm matrix production in V. cholerae and other organisms[49]. To assess intracellular c-di-GMP levels under phospho-DbfR conditions, we introduced an established riboswitch-based c-di-GMP fluorescence reporter into the ΔdbfS mutant[50,51]. Reporter output in this strain was ~40% higher than in WT, confirming that DbfR phosphorylation is associated with elevated intracellular c-di-GMP levels (Fig. 6a). Intracellular c-di-GMP levels are regulated by opposing enzymatic activities: diguanylate cyclases synthesize c-di-GMP via GGDEF domains, while phosphodiesterases degrade c-di-GMP through EAL or HD-GYP domains[52,53]. Of note, V. cholerae encodes 62 such c-di-GMP metabolic enzymes[54], raising the question of which specific enzyme(s) drive the observed c-di-GMP increase upon DbfR phosphorylation. Examination of RNA-sequencing results was not particularly revealing, as widespread changes in the expression of c-di-GMP-metabolizing enzymes were observed in the ΔdbfS mutant (Fig. 6b and Supplementary Data 2).

To pinpoint the critical enzyme(s) responsible for elevated c-di-GMP in the ΔdbfS background, we pursued an unbiased mutagenesis screen that exploits the distinctive colony morphology of the ΔdbfS mutant. This strain exhibits a wrinkled (rugose) colony morphology, a vps-dependent phenotype linked to elevated biofilm matrix production in V. cholerae[55,56], which we use here as a practical phenotypic indicator of hyper-biofilm formation (Fig. 6c). We mutagenized the ΔdbfS strain with the Tn5 transposase, screened ~20,000 colonies, and identified 183 suppressor mutants exhibiting a smooth colony morphology. Sequencing these mutants revealed that 81% harbored disruptions in vps genes (Supplementary Table 2) within the biofilm matrix operons, confirming that the rugose to smooth phenotypic transition reflects loss of biofilm matrix production (Fig. 6d, Left). Each vps gene was disrupted ~8 times on average, validating the comprehensiveness of our screen (Fig. 6d, Middle). We also identified other established biofilm regulators, including the master biofilm transcription factors vpsR and vpsT, the quorum-sensing regulator luxO, and the alternative sigma factor rpoN (Fig. 6d, Right). Most notably, a single c-di-GMP metabolizing enzyme, cdgL (vc_2285), emerged from the screen. This diguanylate cyclase also appeared as the most highly upregulated c-di-GMP metabolizing enzyme in our ΔdbfS transcriptomic dataset (Fig. 6b), suggesting it could mediate elevated c-di-GMP upon DbfR phosphorylation. To test this possibility, we deleted cdgL in the ΔdbfS background, which reduced peak biofilm formation to below that of WT, and restored biofilm dispersal, demonstrating that CdgL is essential for the hyper-biofilm phenotype of the ΔdbfS strain (Fig. 6e). To determine if CdgL links DbfR phosphorylation to elevated c-di-GMP levels, we introduced the c-di-GMP reporter into the ΔdbfS ΔcdgL double mutant. As expected, the ~40% increase in c-di-GMP levels observed in the ΔdbfS strain (Fig. 6a) was nearly abolished in the ΔdbfS ΔcdgL double mutant, which exhibited only a modest (~12%) increase relative to the ΔcdgL single mutant (Fig. 6f). Of note, we found that the ΔcdgL single mutant exhibited a >20% reduction in basal c-di-GMP reporter output and reduced peak biofilm biomass relative to the WT (Supplementary Fig. 8a, b), demonstrating that CdgL also contributes to baseline c-di-GMP synthesis even when DbfR is dephosphorylated. Additionally, the ΔcdgL single mutant exhibited PdbfQRS-lux activity indistinguishable from WT, whereas the ΔdbfS ΔcdgL double mutant exhibited a significant increase relative to WT and was comparable to the ΔdbfS single mutant (Supplementary Fig. 8c). These results indicate that CdgL functions downstream of DbfQRS signaling. Taken together, our results suggest that phosphorylated DbfR activates c-di-GMP synthesis in large part through cdgL upregulation, driving a transcriptional program favoring a sessile biofilm state.

## Constitutive DbfR phosphorylation compromises V. cholerae fitness in an animal model of infection

Given the dramatic transcriptomic changes, reduced growth rate, and commitment to the biofilm state observed upon DbfR phosphorylation, we wondered how activation of this pathway relates to V. cholerae fitness, particularly during infection. To assess fitness, we conducted competition assays in LB media and in the infant mouse model of V. cholerae infection. We competed WT V. cholerae against the dephosphorylated dbfR^D51V strain, as well as the ΔdbfS and ΔdbfQ strains that exhibit elevated DbfR phosphorylation. For both in vitro and in vivo assays, the competitive index (CI) for dbfR^D51V was around 1, indicating that inactivation of DbfR phosphorylation does not impair V. cholerae fitness in media or in the animal (Fig. 7a, b). In contrast, the ΔdbfS and ΔdbfQ mutants were significantly outcompeted by the WT strain under both conditions, with CI values at least 10-fold lower than the dbfR^D51V strain (Fig. 7a, b). Complementation of the ΔdbfS and ΔdbfQ mutants restored fitness in both conditions. We wondered whether the disadvantage of the phospho-DbfR strains was due to their commitment to the biofilm state. To test this possibility, we performed competition experiments

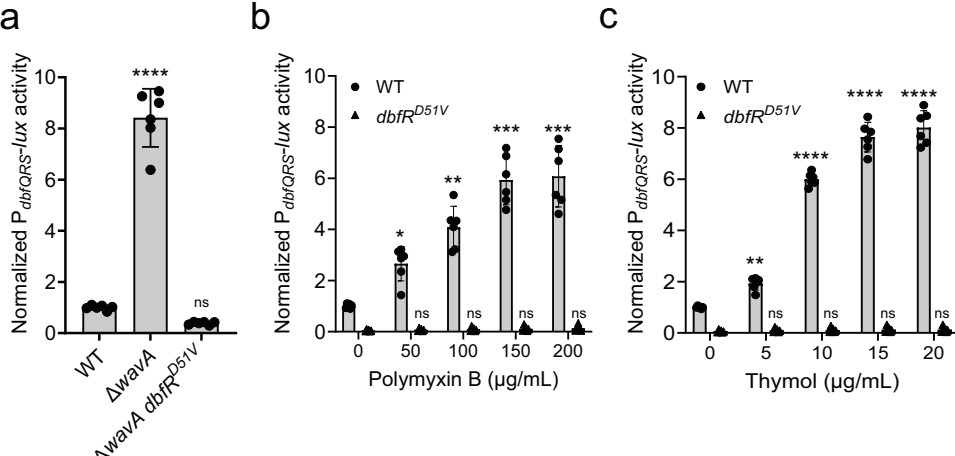

**Fig. 4 | DbfQRS activity is sensitive to membrane disruption. a** Quantification of P*dbfQRS*-*lux* reporter activity for the indicated strains. Data are presented as means ± SD of peak RLU normalized to the average peak value of WT. Points represent individual replicates of $N = 2$ biological replicates and 3 technical replicates. Statistical analysis was performed using one-way ANOVA ($P < 10^{-5}$) with Dunnett's multiple comparisons test to compare each mutant to WT. $P < 10^{-5}$ and $P = 0.2144$ for the indicated pairs. **b** Quantification of P*dbfQRS*-*lux* reporter activity for WT and *dbfR*$^{D51V}$ strains under polymyxin B treatment. Data are presented as mean ± SD of peak RLU normalized to the average peak value of untreated WT. Points represent individual replicates of $N = 2$ biological replicates and 3 technical replicates. Statistical analyses were performed using two-way ANOVA ($P < 10^{-5}$) with Tukey's multiple comparisons test to compare WT and *dbfR*$^{D51V}$ strains under each antibiotic-treated condition to that of the untreated condition. For WT strain, $P = 0.0140, 0.0015, 0.0004$ and $0.00097$ for the indicated pairs; For *dbfR*$^{D51V}$, $P = 0.1251, 0.2854, 0.2482,$ and $0.2968$ for the indicated pairs. $^{*}P < 0.05$; $^{**}P < 0.01$; $^{***}P < 0.001$; $^{****}P < 0.0001$; ns not significant. **c** As in (**b**) for the indicated strains under thymol treatment. For WT strain, $P = 0.0012, <10^{-5}, <10^{-5},$ and $<10^{-5}$ for the indicated pairs; For *dbfR*$^{D51V}$, $P = 0.1494, 0.2171, 0.2526,$ and $0.2709$ for the indicated pairs. $^{*}P < 0.05$; $^{**}P < 0.01$; $^{***}P < 0.001$; $^{****}P < 0.0001$; ns not significant.

with the Δ*dbfS* Δ*cdgL* mutant, which, as shown above, disconnects biofilm regulation from DbfR-phosphorylation. We note that the Δ*dbfS* Δ*cdgL* strain exhibits slower growth than the WT, though its growth defect is less severe than the Δ*dbfS* single mutant (Supplementary Fig. 8d). Consistent with this observation, we found that the Δ*dbfS* Δ*cdgL* strain was also significantly outcompeted in culture and in mice (Fig. 7a, b). Given that 540 genes are differentially expressed in the Δ*dbfS* mutant (Fig. 5a), our results suggest that other components of the regulon, unrelated to the biofilm lifecycle, are responsible for the competitive disadvantage in vitro and in vivo. Overall, our results demonstrate that activation of the DbfQRS pathway negatively impacts the fitness of *V. cholerae* both in culture and in mice. Importantly, our findings highlight that the DbfQRS pathway could serve as a suitable target for the development of novel anti-infectives that function by reducing *V. cholerae* fitness.

## Discussion

The current work represents an initial characterization of a novel class of bacterial TCSs involving a small, secreted protein which directly modulates receptor activity. We provide a comprehensive investigation of the DbfRS TCS in *V. cholerae*—encompassing its input signals, the role of DbfQ as pathway regulator, and downstream effects of DbfR phosphorylation. Using multidisciplinary approaches, we demonstrate that DbfQ controls the activity of DbfS via a direct, high-affinity interaction that is central to pathway regulation. Indeed, the dynamic balance in the opposing kinase and phosphatase activities of DbfS calibrates the phosphorylation state of DbfR, which in turn determines *V. cholerae*'s decision to disperse or commit to a biofilm state. We reveal that defects in LPS biosynthesis or the presence of membrane-targeting agents elevate DbfR phosphorylation, implicating the pathway in responding to outer membrane perturbations. Although the activating cue(s) for DbfQRS remain unknown, we suggest two possibilities that could explain its response to membrane damage. (1) DbfQ or DbfS could directly sense a cell-intrinsic feature of membrane damage, or (2) increased outer-membrane permeability could allow regulatory signal(s) to enter or escape from the periplasm. In either

case, these signals could modulate the affinity of the DbfQ–DbfS interaction, and in turn, DbfR phosphorylation. Future investigations will be required to resolve these hypotheses.

The DbfQRS pathway exerts pleiotropic control over the biofilm lifecycle, metabolic activities, and motility, underscoring its wide-ranging influences on *V. cholerae* physiology. Strikingly, constitutive activation of the pathway imposes a fitness cost, compromising *V. cholerae* growth and colonization. Consistent with this finding, we note that a previous attempt to generate an in-frame deletion of *dbfS* in a different *V. cholerae* isolate was unsuccessful, suggesting that constitutive DbfR activation may even be lethal in some strain backgrounds[57]. In the current study, we find that the growth defect observed upon pathway activation is not due to the costs associated with elevated biofilm production. Thus, we propose that the fitness cost of DbfQRS hyperactivity is likely to arise from transcriptional downregulation of metabolic processes. Pinpointing the pathways controlled by DbfR that impact growth will be the subject of future studies.

Despite the costs of DbfQRS hyperactivation, the widespread conservation of this pathway suggests it confers a selective advantage under specific conditions. Given that DbfQRS responds to outer membrane stresses, its activation may serve as a regulatory mechanism that prioritizes stress adaptation over cellular activity. By reprogramming cellular processes (i.e., enhancing biofilm formation while downregulating metabolism, motility, and growth), DbfQRS activation likely benefits *V. cholerae* in hostile environments where survival takes precedence over rapid proliferation. For example, downregulation of metabolic processes may conserve energy during nutrient deprivation while biofilm formation could provide protection against environmental stressors such as predation or antimicrobials. The DbfQRS pathway is distinct in that, unlike many TCSs that regulate specific aspects of bacterial adaptation, it orchestrates a broad, pleiotropic shift in cellular physiology—encompassing biofilm regulation, metabolism, motility, and cell envelope integrity. Such expansive regulatory control underscores its unique role as a master switch for multiple adaptation strategies in adverse environments.

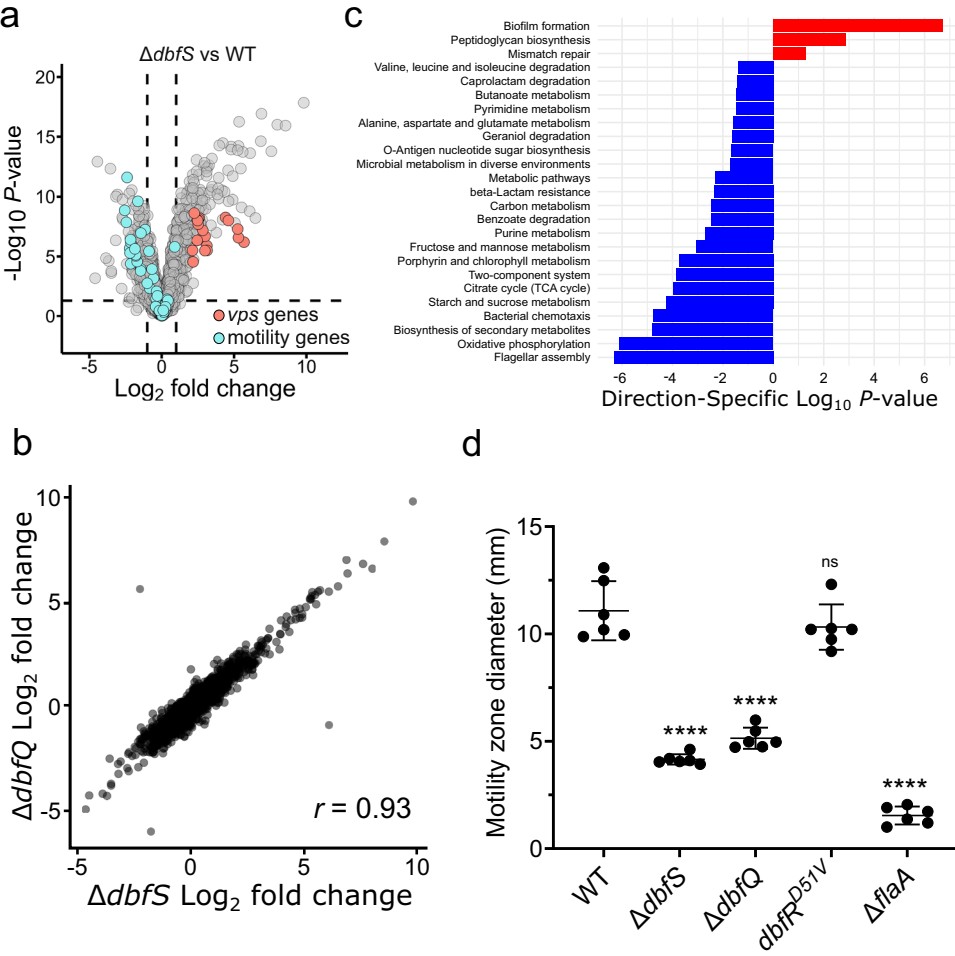

**Fig. 5 | The DbfQRS pathway regulates the expression of metabolic and lifestyle genes. a** Volcano plot comparing fold changes and *P*-values for gene expression in the Δ*dbfS V. cholerae* strain compared to WT. *ups* and motility genes are highlighted in orange and cyan, respectively. The horizontal dotted line represents a -Log$_{10}$ *P*-value of 0.05, and left and right vertical dashed lines represent Log$_2$ fold changes of −1 and 1, respectively. Differential expressions were assessed using edgeR's Quasi-Likelihood F-test (two-sided). *P*-values were adjusted for multiple comparisons using the Benjamini−Hochberg false discovery rate (FDR). Samples were collected at OD$_{600}$ = 0.1 and *N* = 3 biological replicates for each strain. Complete datasets are available in Supplementary Data 2. **b** Comparison of Log$_2$ fold changes in gene expression for the Δ*dbfS* and Δ*dbfQ* mutants. *r* = Pearson's correlation coefficient. Complete datasets are available in Supplementary Data 2 and 3. **c** Significantly enriched KEGG pathways (*P* < 0.05) in the Δ*dbfS* mutant strain. Differentially regulated genes (false discovery rate <0.05) were assigned to pathways using the limma's *kegga* function[73] with default parameters. Enrichment *P*-values were two-sided and adjusted for multiple comparisons using the Benjamini−Hochberg FDR. Red bars represent upregulated pathways, and blue bars represent downregulated pathways. **d** Motility zone measurement for the indicated strains after 10 h of incubation. Points represent individual replicates of *N* = 6 biological replicates, ± SD. Statistical analysis was performed using unpaired, two-sided t-tests with a 95% confidence interval to compare each mutant to WT. *P* < 10$^{-5}$, <10$^{-5}$, = 0.3097, and <10$^{-5}$ for the indicated pairs. ****P < 0.0001; ns not significant.

In summary, our work reveals that the DbfQRS pathway integrates membrane stress signals with global regulatory responses, thereby linking environmental challenges to adaptive lifestyle transitions in *V. cholerae*. More broadly, understanding how stress-responsive TCSs like DbfQRS orchestrate bacterial adaptation may reveal novel antimicrobial strategies that exploit the inherent trade-offs in bacterial stress adaptation. Our findings present the DbfQ−DbfS interface as a potential site for the development of small-molecule modulators. Agents that directly disrupt DbfQ−DbfS binding and bias DbfS toward kinase output would hyperactivate the pathway, leading to a low-fitness state with reduced growth and colonization. Given the widespread conservation of *dbfQRS*-like modules across bacterial phyla, it is likely that similar regulatory principles extend beyond *V. cholerae*. Future studies should explore whether this pathway plays a comparable role in other clinically and environmentally relevant bacteria. Examining the function of DbfQRS homologs in diverse species could reveal conserved and species-specific regulatory strategies, offering insight into how different bacteria fine-tune stress responses.

Additionally, structural and biochemical characterization of DbfQ orthologs could determine whether their mode of receptor modulation is universally conserved or has evolved to accommodate distinct signaling contexts. A deeper understanding of the DbfQRS family may not only refine our current models of bacterial signaling networks but also uncover new targets for antimicrobial interventions by disrupting stress adaptation mechanisms in pathogenic bacteria.

## Methods
### Bacterial strains, reagents, and cloning
The *V. cholerae* parent strain used in this study was O1 El Tor biotype C6706str2. *E. coli* S17 and Top10 were used to transfer plasmids into *V. cholerae* by conjugation. A complete list of strains used in this study is provided in Supplementary Data 5. For passaging and cloning, *V. cholerae* and *E. coli* strains were cultivated in lysogeny broth (LB) supplemented with 1.5% agar or in liquid LB with shaking at 30 °C and 37 °C, respectively. Unless otherwise specified, antibiotics were used at the following concentrations: polymyxin B, 50 μg/mL;

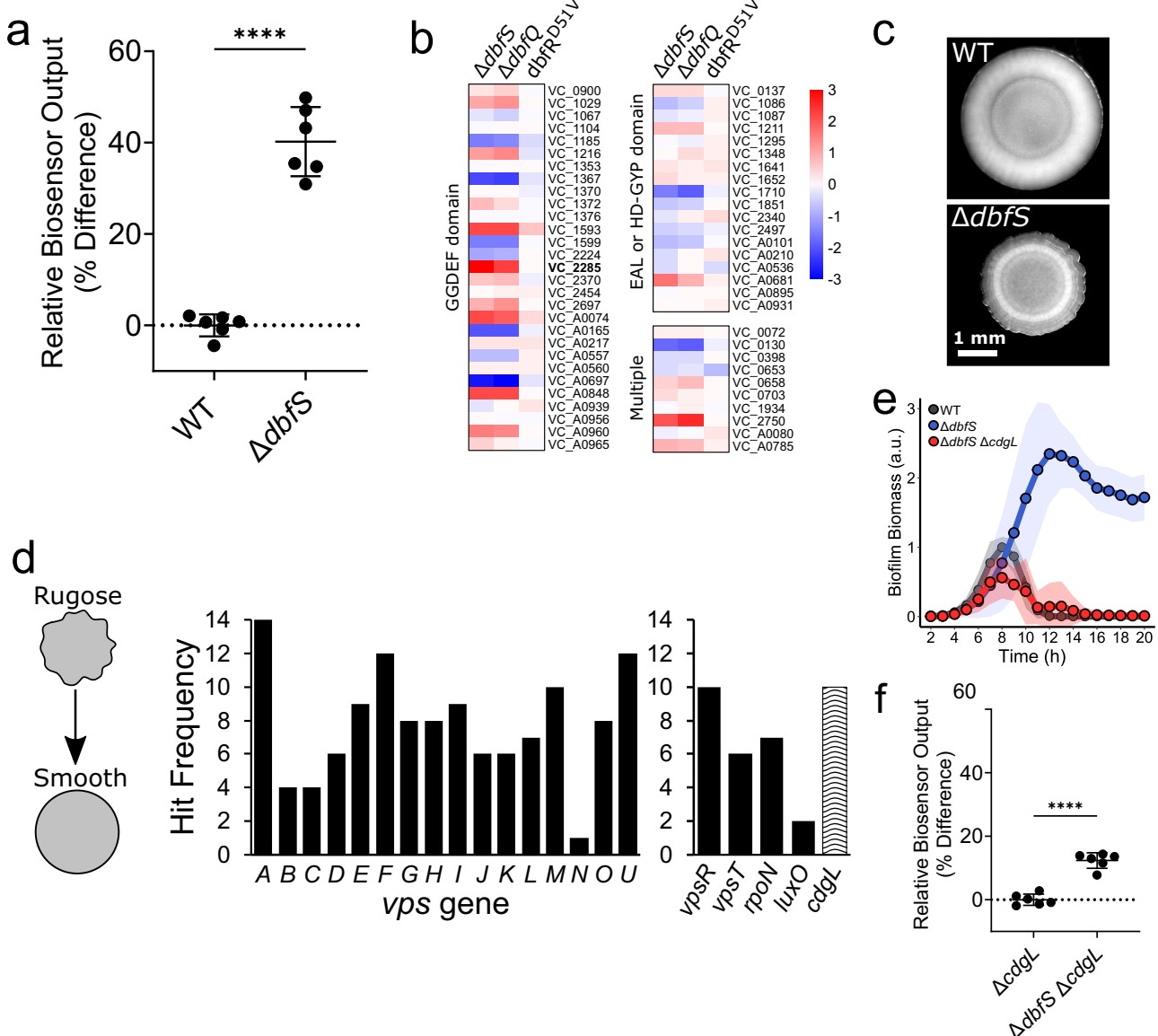

**Fig. 6 | DbfR phosphorylation drives increased c-di-GMP levels via upregulation of *cdgL*. a** Relative c-di-GMP reporter output for the WT and Δ*dbfS* strains, expressed as means ± SD of percentage difference relative to the WT control. Points represent individual replicates of *N* = 2 biological replicates and 3 technical replicates. Statistical analysis was performed using unpaired, two-sided t-tests with a 95% confidence interval. *P* < 10⁻⁵. **b** Heatmap of RNAseq results for c-di-GMP metabolizing enzymes in the indicated strains. Genes are grouped by the presence of GGDEF domain (diguanylate cyclase), EAL or HD-GYP domains (phosphodiesterase), or multiple catalytic domains, potentially representing bifunctionality. Color bar represents Log₂ fold changes. *VC_2285* (*cdgL*) is bolded for ease of identification. **c** Stereomicroscope images of WT and Δ*dbfS* colony morphologies. Data are representative of *N* = 3 biological replicates. **d** Left: Schematic of mutagenesis screen. Middle: Hit frequency for genes in the *vps-I* and *vps-II* operons. Right panel: Hit frequency for selected regulators of biofilm formation. All screen hits are reported in Supplementary Table 2. **e** Quantification of biofilm biomass for WT, Δ*dbfS*, and Δ*dbfS* Δ*cdgL* strains using time-lapse brightfield microscopy. Points represent means ± SD (shaded regions) of *N* = 2 biological replicates and 3 technical replicates. **f** As in (**a**) for the Δ*cdgL* and Δ*dbfS* Δ*cdgL* double mutant. Reporter output is normalized to the Δ*cdgL* strain. *P* < 10⁻⁵. ****P < 0.0001. a.u. arb. units.

kanamycin, 50 µg/mL; spectinomycin, 200 µg/mL; streptomycin, 400 µg/mL, chloramphenicol, 2 µg/mL; gentamicin, 15 µg/mL; and ampicillin, 100 µg/mL. For Pho-Lac fusion experiments, cultures were spotted onto LB plates supplemented with 80 µg/mL Red-Gal (6-chloro-3-indolyl-β-D-galactoside) and 100 µg/mL X-Pho (5-bromo-4-chloro-3-indolyl-phosphate). For microscopy, luminescence quantifications, and c-di-GMP reporter measurements, *V. cholerae* strains were grown in M9 minimal medium supplemented with dextrose and casamino acids (1x M9 salts, 100 µM CaCl₂, 2 mM MgSO₄, 0.5% dextrose, 0.5% casamino acids). When required, *L*-arabinose (Thermo Fisher Scientific) was added at a concentration of 0.2% from the start of assays.

All genetic modifications were generated by replacing genomic DNA with linear DNA introduced via natural transformation, as described previously[58,59]. PCR amplification was performed with iProof (Bio-Rad) or Q5 DNA polymerase (New England Biolabs). Sanger sequencing (Azenta) was used to verify genetic alterations. Genomic DNA from recombinant strains was used as templates for generating DNA fragments as needed. Gene deletions were generated in-frame to remove the entire coding sequences, except for *dbfS* and *dbfR*, which overlap in their operon. For these genes, an internal portion of each gene was deleted to avoid perturbing the adjacent gene. To construct *dbfQ-3×FLAG* at its native locus, a C-terminal 3×FLAG epitope tag was inserted immediately upstream of the *dbfQ* stop codon.

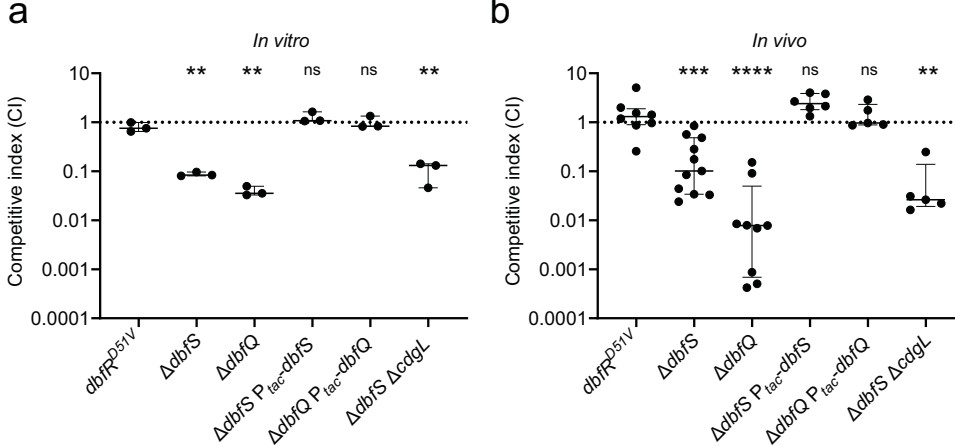

Fig. 7 | **Activation of the DbfQRS pathway results in a competitive disadvantage in vitro and in vivo. a** In vitro competition assay. WT (*lacZ⁻*) *V. cholerae* were competed against the indicated strains at a 1:1 ratio in LB medium for 24 h. Data are displayed as medians with interquartile ranges. Points represent individual replicates of $N = 3$ biological replicates. Statistical analyses were performed using unpaired, two-sided t-tests with a 95% confidence interval. $P = 0.0021$, 0.0016, 0.0977, 0.3696, and 0.0027 for the indicated pairs. **b** In vivo competitive colonization assay. WT (*lacZ⁻*) was competed against the indicated strains at a 1:1 ratio in the infant mouse small intestine for 24 h. Data are presented as a single data point per mouse, along with median and interquartile range for each condition. Statistical analyses were performed using two-sided Mann-Whitney U test. $P = 0.0003$, 0.00008, 0.0813, 0.8329, and 0.0016 for the indicated pairs. In both assays, competitive indices (CI) were calculated as the ratio of output to input of the mutant strain relative to the WT. *$P < 0.05$; **$P < 0.01$; ***$P < 0.001$; ****$P < 0.0001$; ns not significant.

Oligonucleotides and synthetic linear DNA g-blocks were ordered from IDT and are reported in Supplementary Data 6.

## Microscopy and image analysis

The *V. cholerae* biofilm lifecycle was monitored by time-lapse brightfield microscopy as described previously[60]. Briefly, single colonies of *V. cholerae* were inoculated into 200 μl of LB medium in 96-well plates covered with a breathe-easier membrane (USA Scientific Inc.) and grown overnight at 30 °C with constant shaking. The following morning, overnight cultures were diluted ~1:200,000 in M9 minimal medium to achieve a final $OD_{600}$ of ~$10^{-5}$. Diluted cultures were statically grown at 30 °C in 96-well polystyrene microtiter plates (Corning) in an Agilent Biospa robotic incubator which transferred microtiter plates for brightfield imaging at 30-min intervals over a 24-h period. Image acquisition was performed using the Agilent Biotek Cytation 1 imaging plate reader equipped with a 10x air objective (Olympus Plan Fluorite, NA 0.3) or a 4x air objective (Olympus Plan Fluorite, NA 0.13), controlled by the Biotek Gen5 (version 3.12) software. Quantifications of biofilm biomass were performed based on the principle that biofilms scatter light to a greater degree than planktonic cells in low-magnification brightfield images. To segment biofilms within brightfield images, first pixel intensities were inverted, local contrast was normalized, images were blurred with a Gaussian filter, and a fixed threshold was applied, allowing for the differentiation of biofilms from background. The resulting binarized image masks were then applied to the raw images to determine the total light attenuated by biofilms within the field of view, yielding our biofilm biomass metric. Peak biofilm biomass corresponding to the maximum value recorded in each time-lapse replicate. In all cases, biomass values were normalized to the peak biomass of the control strain to account for day-to-day variability and for ease of comparison between strains/conditions. Image analyses were performed using the Julia programming language (version 1.11.1)[60]. Plotting was performed in RStudio (version 4.4.1)[61] using the ggplot2 package. Figures and original cartoons were assembled using Inkscape software (version 1.4).

Fluorescence microscopy used to investigate DbfQ localization was performed on a DMI8 Leica SP-8 point scanning confocal microscope driven by LasX software (version 3.7). For imaging, strains harboring DbfQ fusions to mNG were first grown in M9 medium to $OD_{600} = 0.6$, at which point cultures were diluted 1000x and inoculated into glass-bottomed 96-well plates (Mattek). Cells were then allowed to attach for 1 h before imaging. Microscopy was performed with a 63x objective (Leica, NA = 1.20), along with a digital zoom of 5x. A tunable white-light laser (Leica; model #WLL2; excitation window = 470–670 nm) set to 503 nm was used to excite mNG fluorescence. Light was detected using GaAsP spectral detectors (Leica, HyD SP), and timed gate detection and frame averaging were employed to minimize background signal.

## Gene neighborhood analysis

To assess if PepSY domain-containing proteins (PDPs) are frequently found in the genomic proximity of histidine kinases (HKs) and/or response regulators (RRs) in bacterial genomes, we performed a non-comprehensive screen using a large database of proteins from InterPro[62]. First, we downloaded gene IDs and organism IDs for three sets of proteins: (1) ~1.2 M proteins containing the HK domain (IPR005467), (2) ~1.6 M proteins containing the RR domain (IPR001789), and (3) ~17 K proteins containing only the PepSY domain (IPR025711). Gene IDs for each protein were of the form "<genome-ID>_<gene-number>". Second, using a custom Python script, we parsed the above gene IDs and identified all pairs of PDP and HK/RR genes that lie within 4-gene numbers in the same genome. This stringent cutoff was not selected to be comprehensive, but instead to reveal genomes where three gene classes are closely positioned on the genome. Third, to allow an assessment of our result, we estimated the null probability of a PDP lying near an HK/RR by assuming an average circular bacterial genome of 4000 genes, and one copy each of a PDP and an HK/RR distributed randomly in this genome. The observed frequency of a PDP and an HK/RR in proximity was found to be >60-fold higher than null.

To identify which taxa had a PDP and an HK/RR within a 4-gene vicinity, we used organism IDs from the shortlisted set above and determined their taxonomic order, class and phylum using the tree of life found in the STRING database (string-db.org/cgi/download)[63]. The order Vibrionales (containing genus *Vibrio*) made up 0.8% of the dataset. Therefore, all taxa with a representation of ≥0.8% were shown in the table (Fig. S2). Finally, for a handpicked set of species relevant to human health and agriculture, gene

neighborhood data were displayed (Fig. 1e) using the FlaGs pipeline[64] with the following RefSeq IDs of PDPs as input and default parameters: WP_001165432.1, WP_011080671.1, WP_003090497.1, WP_000780501.1, WP_004201003.1, and WP_003592804.1.

## Phos-tag gel analysis

To monitor DbfR and phospho-DbfR via SDS-PAGE, phos-tag gel experiment was carried out as described previously[23], with some minor adjustments. Briefly, the endogenous *dbfR* gene was replaced with *dbfR-SNAP* in Δ*dbfS* and Δ*dbfQ* Δ*dbfS* strains, and an arabinose-inducible $P_{BAD}$-*dbfS* was introduced at an ectopic locus (*vc_1807*). Overnight cultures of each strain were diluted 1:1,000 in 3 mL of LB and grown at 30 °C with shaking to an $OD_{600}$ of ~0.6, after which 1 μM SNAP-Cell TMR Star (New England Biolabs) was added to label the SNAP tag. Cultures were subsequently divided into two tubes: 0.2% *L*-arabinose was added to one tube to induce DbfS production, while the other tube was left uninduced. The cultures were returned to 30 °C with shaking and after 1-h incubation, 1 mL of the cultures were harvested by centrifugation ($18,000 \times g$, 1 min). Pelleted cells were subsequently lysed in 40 μL of Bug Buster (EMD Millipore, #70584-4) for 5 min at 25 °C with intermittent vortexing. Lysates were solubilized in 1.5x SDS/PAGE buffer for 5 min, and samples were immediately loaded onto a cold 7.5% SuperSep Phos-tag gel (50 μM) (FUJIFILM Wako Pure Chemical; 198-17981). Electrophoresis was carried out at 80 V at 4 °C for ~1 h. Gel images were captured on the FluorChem E Imaging System (ProteinSimple).

## Quantification of $P_{dbfQRS}$-*lux*

To quantify the transcriptional activity of the *dbfQRS* promoter as a proxy for DbfR phosphorylation, we constructed a $P_{dbfQRS}$-*lux* reporter by fusing the *dbfQRS* promoter to the *luxCDABE* luciferase operon from *Photorhabdus*[65]. All strains used in this assay harbored a *upsL* deletion to eliminate biofilm-related interference with luminescence and optical density measurements. Unless indicated otherwise, overnight cultures of strains harboring the $P_{dbfQRS}$-*lux* reporter were diluted to $OD_{600}$ of ~$10^{-5}$ in fresh M9 medium in 96-well plates. Optical density at 600 nm ($OD_{600}$) and luminescence intensity (*lux*) from $P_{dbfQRS}$-*lux* were measured simultaneously at 30-min or 1-h time intervals over a 24-h period using the Agilent Biotek Cytation 1 Plate Reader, driven by the Biotek Gen5 software. For quantification of $P_{dbfQRS}$-*lux* reporter output in the presence of antimicrobials or iron, cultures were adjusted to a starting $OD_{600}$~0.1 at the onset of assay. For antimicrobial assay, overnight cultures were diluted in M9 medium supplemented with increasing concentrations of polymyxin B or thymol. For iron supplementation assay, overnight cultures were diluted in M9 medium supplemented with 100 μM $FeCl_2$ or $FeCl_3$, followed by a shortened 2-h assay to minimize Fe(II) oxidation.

## Transposon mutagenesis screens

Two independent Tn5 transposon mutagenesis screens were performed: (1) to identify mutations activating $P_{dbfQRS}$-*lux*, and (2) to identify suppressors of colony rugosity in the Δ*dbfS* background. In both cases, transposon mutagenesis was carried out identically: *E coli* strain S17 harboring the Tn5::*kan* transposase plasmid was conjugated with the specified parent *V. cholerae* strains. Conjugation was allowed to proceed for two hours at 37 °C on LB agar plates without selection. The short incubation ensured the limited appearance of sister mutants derived from cell divisions following transposon insertion. Conjugations were collected with a sterile loop, resuspended in liquid LB media, and plated on selective media. Transposon insertion mutants were selected on LB agar plates supplemented with streptomycin (to kill *E. coli*) and kanamycin (to select for transposon integration). For the $P_{dbfQRS}$-*lux* screen, mutant colonies with visibly increased luminescence, as observed on a FluorChem E imager, were isolated, grown overnight, and arrayed into 96-well plates for quantification of $P_{dbfQRS}$-

*lux* reporter activity as described above. Mutants exhibiting significantly elevated luminescence compared to WT control were selected for further analysis. In the Δ*dbfS* colony rugosity suppressor screen, mutants displaying a smooth colony morphology (rugose-to-smooth transition) were identified by visual inspection and selected for sequencing. The locations of transposon insertions in both screens were determined using arbitrary PCR (Supplementary Data 6), followed by Sanger sequencing (Azenta).

## Western blotting

Cultures of strains expressing the DbfQ-3×FLAG, DbfS-3×FLAG, and their relevant variants were grown to $OD_{600} = 1.0$, then harvested by centrifugation for 1 min at $18,000 \times g$. The resulting pellets were flash frozen, thawed, and lysed for 10 min at 25 °C by resuspension to $OD_{600} = 1.0$ in Bug Buster (EMD Millipore) supplemented with 0.5% Triton-X, 50 μg/mL lysozyme, 25 U/mL benzonase nuclease, and 1 mM phenylmethylsulfonyl fluoride (PMSF). 1x SDS-PAGE buffer (final) was then added and allowed to incubate for 1 h at 37 °C before loading into 4–20% Mini-Protein TGX gels (Bio-Rad). Electrophoresis was performed at 200 V for 30 min. Protein transfer onto PVDF membranes (Bio-Rad) was then performed for 1 h at 4 °C at 100 V in transfer buffer (25 mM Tris, 190 mM glycine, 20% methanol). Membranes were blocked for 1 h in 5% milk in PBST (137 mM NaCl, 2.7 mM KCl, 8 mM $Na_2HPO_4$, 2 mM $KH_2PO_4$, and 0.1% Tween). Membranes were subsequently probed for 1 h using a monoclonal Anti-FLAG-Peroxidase antibody (Millipore Sigma, #A8592) at a 1:5000 dilution in PBS-T containing 5% milk. After 6, 5-min washes in PBST, membranes were developed using the Amersham ECL western blotting detection reagent (GE Healthcare). For the RpoA loading control, the same protocol was followed except that the primary antibody was Anti-*Escherichia coli* RNA Polymerase α (Biolegend, #663104) used at a 1:10,000 dilution and the secondary antibody was an Anti-Mouse IgG HRP conjugated antibody (Promega, #W4021) used at a 1:10,000 dilution.

## AlphaFold structure prediction and interface analysis

Protein sequences for PepSY domain-containing protein(s) (DbfQ-like) and their cognate histidine kinase (DbfS-like) from *V. cholerae* and *P. aeruginosa* were submitted to the AlphaFold Server (AlphaFold3[34]; https://alphafoldserver.com/) using default parameters for hetero-complex prediction. For each run, five models were generated, and the highest-ranked model based on server-reported confidence scores was selected for further visualization and analysis. Predicted protein 3D structures were visualized in PyMOL[66] (version 3.1, Schrödinger, LLC). Putative electrostatic contacts at the DbfQ−DbfS interface were identified as salt bridges between oppositely charged side-chain atoms within ≤4.0 Å, using PyMOL's "Find → salt bridge interaction" tool (Action menu), and verified by manual inspection. All residue numbers refer to the full-length protein sequence of DbfQ or DbfS.

## Purification and pull-down assays for DbfQ and DbfS$^{SD}$

DNA encoding processed DbfQ-6×His (residues 32 to 135, excluding the secretion signal), and DbfS$^{SD}$-6×His (residues 43 to 171) were cloned into the pET-15b vector via Gibson assembly (NEB), followed by transformation into chemically competent *E. coli* BL21 (DE3) cells. For protein purification, cultures were grown in Terrific broth (Fisher BioReagents) supplemented with 100 μg/mL ampicillin at 37 °C for 4 h with shaking until $OD_{600}$~0.6−0.8 was achieved. Protein expression was induced with 1 mM IPTG, followed by incubation at 18 °C for 16 h with continuous shaking. The following morning, cells were harvested by centrifugation ($5000 \times g$, 20 min, 4 °C) and resuspended in lysis buffer (25 mM Tris−HCl pH 7.5, 300 mM NaCl, 10 mM imidazole, 10% glycerol, 0.5 mg/mL lysozyme, 25 U/mL benzonase nuclease, and 1x EDTA-free protease inhibitor tablet (Thermo Scientific). To stimulate lysis, resuspended cells were subjected to sonication (4 min total; 30 s on, 1:30 off; 8x cycles). Lysates were then subjected to centrifugation

(25,000 × $g$, 20 min, 4 °C). Supernatants were filtered through 0.45 μm filters (Cytiva, #4654) and loaded onto Ni-NTA resin (EMD Millipore, #70666-4) pre-equilibrated with lysis buffer. After lysate flow-through was completed, columns were washed 3 times with 10x column volumes of wash buffer (25 mM Tris–HCl pH 7.5, 300 mM NaCl, 20 mM imidazole, 10% glycerol), after which bound proteins were eluted with elution buffer (25 mM Tris–HCl pH 7.5, 100 mM NaCl, 300 mM imidazole, 10% glycerol). The 6×His tag on DbfQ was removed by thrombin cleavage (Cytiva Thrombin Protease, #45001320), and cleaved DbfQ was separated from the His-tag and uncleaved protein by repeated Ni-NTA affinity chromatography. Proteins were then dialyzed into storage buffer (25 mM Tris–HCl pH 7.5, 100 mM NaCl) using 3500 molecular weight cut-off dialysis cassettes (Thermo Scientific, #66330) overnight at 4 °C. Protein purity and concentration were assessed by SDS-PAGE and Pierce BCA Protein Assay (Thermo Scientific, #23225), respectively. Aliquots of proteins were flash-frozen in liquid nitrogen and stored at −80 °C for subsequent use.

For pull-down assays, 50 μM of DbfQ (6×His tag-cleaved) and 50 μM of DbfS$^{SD}$-6×His were mixed in binding buffer (25 mM Tris–HCl pH 7.5, 100 mM NaCl, 1x EDTA-free protease inhibitor tablet). The protein mixture was incubated for 1 h at 25 °C, then applied to Ni-NTA resin. After three washes with wash buffer, bound proteins were eluted with elution buffer and analyzed by SDS-PAGE. As controls, individual DbfQ (6×His tag-cleaved) and DbfS$^{SD}$-6×His were incubated with Ni-NTA resin under identical conditions.

## Microscale thermophoresis (MST) analysis

Experiments were performed as described previously with minor modifications[67]. Before labeling with fluorescent probes, DbfQ was diluted and buffer exchanged into buffer M (20 mM MES, 100 mM NaCl, 10% glycerol) to remove incompatible buffer components. For protein labeling, 10 μM DbfQ was mixed with 3x excess Red NHS dye (Nanotemper Technologies) dissolved in DMSO and incubated at room temperature in the dark for 30 min. After incubation, excess and unreacted dye was removed by passing the protein-dye mixture through a gel-filtration column (Column B, Nanotemper Technologies). Optimal protein labeling was determined using the formula: $A_{650}/195,000/M/cm$ x concentration of labeled protein; $A_{650}$ = absorbance at 650 nm, and 195,000/M/cm is the molar absorbance of the Red NHS dye. An optimal labeling was considered as a value between 0.6 and 1.

For MST experiments, unlabeled DbfS protein was serially diluted in low binding tubes in buffer M supplemented with 0.05% Tween 20 at pH 7.5. The labeled DbfQ protein was added and incubated for 5 min at room temperature. The concentration of unlabeled DbfS ranged from 0.49 nM to 16,105 nM while the concentration of the labeled DbfQ was kept constant (20 nM). After incubation, samples were loaded into standard Monolith NT.115 capillary tubes and thermophoresis was determined using the Monolith MST device (Nanotemper Technologies) with the following parameters: 20–60% excitation power and medium MST Power at 25 °C. Thermophoresis results were analyzed using the PALMIST[68] and GUSSI[69] analysis pipeline. Briefly, data from Monolith Software were imported into the PALMIST software and a preset T-jump (TJ) was applied to the data using a 1:1 binding model with 95% confidence interval. After data analysis, figures were rendered using GUSSI. For stoichiometry determination, DbfQ-NHS (100 nM) was titrated against narrow increasing concentrations of unlabeled DbfS (0 nM to 40,000 nM) using 20–60% excitation power and medium MST Power at 25 °C. Fluorescence values were converted into relative thermophoresis by dividing the normalized fluorescence by the resulting amplitude at each data point.

## RNA-sequencing

Cultures of the indicated *V. cholerae* strains, grown in triplicate, were diluted to $OD_{600}$-0.001 in 5 mL of M9 medium. Subcultures were incubated at 30 °C with shaking until $OD_{600}$ 0.1 was reached. At this point, cells were collected by centrifugation for 10 min at 3200 × $g$ and resuspended in RNAprotect (Qiagen). RNeasy mini kit (Qiagen) was used for RNA isolation and a TURBO DNA-free kit (Invitrogen) was used to remove remaining DNA. The concentration and purity of RNA were measured using a NanoDrop instrument (Thermo). Samples were frozen in liquid nitrogen and stored at −80 °C until they were shipped on dry ice to the Microbial Genome Sequencing Center (now SeqCoast). Upon sample submission, the 12 million paired-end reads option and the intermediate analysis package were selected for each sample. As per the MIGS project report, quality control and adapter trimming were performed with bcl2fastq (Illumina), while read mapping was performed with HISAT2[70]. Read quantitation was performed using Subread's featureCounts[71] functionality, and subsequently, counts were loaded into R (R Core Team) and normalized using edgeR's[72] Trimmed Mean of M values (TMM) algorithm. Values were converted to counts per million (cpm), and differential expression analyses were performed using edgeR's QuasiLinear F-Test (qlfTest) functionality against treatment groups, as indicated. Kegg pathway analysis was performed using limma's[73] "kegga" functionality with default parameters. Genes considered Up/Down in this analysis had a false discovery rate < 0.05. Plots, including heatmaps, pathway regulation, and volcano plots, were produced in RStudio using the ggplot2 package.

## Motility assays

Motility assays were performed on 6-well plates containing 1% tryptone, 0.5% NaCl, and 0.3% agar. *V. cholerae* strains were first grown overnight on LB agar plates at 30 °C. A single colony from each strain was picked by a sterile pipette tip and inoculated into the center of each well of the motility plates. Plates were incubated at 37 °C for 16 h, during which images were taken every hour using an Epson V750 Pro flatbed scanner. Motility was quantified by measuring the diameter of the motility zone for each strain in Fiji (version 1.54k)[74]. All statistical analyses in this study were performed using GraphPad Prism version 10.4.1 (GraphPad Software, San Diego, CA, USA).

To assess single-cell swimming motility, bacterial strains were cultured overnight in LB medium. The following day, cultures were diluted to an optical density of OD = 0.001 in M9 medium. Aliquots of 200 μL were dispensed into glass-bottom 96-well plates (MatTek) and incubated for 1 h at room temperature to allow cells to attach to the coverslip surface. Wells were then washed three times with fresh medium to remove non-adherent cells. Plates were incubated at 30 °C for an additional 3 h prior to imaging, such that cells could detach and enter the motile state. Brightfield time-lapse microscopy was performed on a Biotek Cytation 5 microscope at 20x magnification with a frame interval of 20 ms, capturing cells at a depth of -100 μm into the sample. Raw image sequences were preprocessed by smoothing and background subtraction, and trajectories were extracted using the TrackMate[75] plugin in Fiji. Cell detection was carried out with a Laplacian of Gaussian filter, and tracks were linked using the simple linear assignment problem (LAP) algorithm. Resulting trajectory data were imported into R for downstream analysis. Median swimming speeds for each trajectory were extracted and the number of trajectories per strain was capped at $N = 225$ by randomly subsampling when necessary.

## c-di-GMP reporter assay

The c-di-GMP reporter assays were performed as previously described[50]. Strains were constructed by conjugation with *E. coli* Top10 and S17 cells. Briefly, overnight *V. cholerae* cultures were diluted 1:5000 into M9 medium supplemented with 15 μg/mL gentamicin and transferred into 96-well plates sealed with a breathe-easier membrane. Plates were incubated overnight at 37 °C with shaking. The following day, the membrane was removed and reporter measurements were obtained using a Tecan Spark plate reader using AmCyan (ex: 430 ± 20 nm, em: 490 ± 20 nm) and Turbo RFP (ex: 520 ± 20 nm, em: 580 ± 20 nm) channels. Relative fluorescence intensity (RFI) was

calculated as the c-di-GMP-regulated TurboRFP signal divided by constitutive AmCyan signal, and RFI values of mutants were normalized to WT signal.

## In vitro and in vivo competition assays

Competition assays were conducted to assess fitness of mutant *V. cholerae* strains relative to a WT reference strain (*lacZ*−) both in LB medium (in vitro) and in an infant mouse model (in vivo). For both assays, bacterial cultures were grown aerobically for ~18 h in LB medium at 30 °C. For in vitro assays, ~$10^5$ CFU of each competing strain were mixed 1:1 in LB medium containing 4 mm glass beads (to disrupt biofilms, as has been done previously) and incubated for 24 h at 30 °C. At the onset of the experiment and after 24 h of competition, samples were collected, serially diluted, and plated on LB/X-gal plates to enumerate CFUs for each strain. For in vivo competition assays, competing strains were mixed equally at a 1:1 ratio, and approximately $10^6$ CFU were administered orogastrically to 4 to 7-day-old CD-1 mice (Charles River Laboratories). Prior to infection, infant mice were housed with their dam with ample access to food and water for at least 24 h and monitored. Serial dilutions of the intestinal homogenates were plated on LB/X-gal agar plates, allowing for CFU enumeration. In both assays, competitive index (CI) was calculated as the ratio of output to input of the mutant strain relative to the WT. For in vitro assays, three biological replicates were performed, while a minimum of five mice were used for each in vivo experiment. Each biological replicate (in vitro) or individual mouse (in vivo) was treated as a single data point, and results are presented as the median with interquartile range. The sex of neonatal CD-1 mice was not determined at the time of use. Both sexes were randomly included in experiments. Sex-based analyses were not performed as there are no known sex-dependent differences in this infant mouse colonization model. All animal experiments were approved by the Institutional Animal Care and Use Committee at Tufts University School of Medicine (Protocol B2024-26).

## Ethics statement

All animal experiments were done in accordance with NIH guidelines, the Animal Welfare Act, and US federal law. The infant mouse colonization experimental protocol B2024-26 was approved by Tufts University School of Medicine's Institutional Animal Care and Use Committee. All animals were housed in a centralized and AAALAC-accredited research animal facility that is fully staffed with trained husbandry, technical, and veterinary personnel in accordance with the regulations of the Comparative Medicine Services at Tufts University School of Medicine.

## Reporting summary

Further information on research design is available in the Nature Portfolio Reporting Summary linked to this article.

# Data availability

The source data used to generate all main and supporting figures in this work are available on Figshare (https://doi.org/10.1184/R1/28653392) or as Supplementary Data files. Transcriptomic results are available in ArrayExpress (E-MTAB-16165).

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

## Acknowledgements

We thank members of the Bridges lab for their insightful discussions and detailed feedback on the manuscript. We especially thank Dr. Bonnie Basler for guidance and for sharing strains. This work was supported by NIH grant R00AI158939, NIGMS grant 1R35GM160020, a Shurl and Kay Curci Foundation grant (https://curcifoundation.org/), a Kaufman Foundation New Investigator Research Grant KA2023-136488 (https://kaufman.pittsburghfoundation.org/), a Damon Runyon Cancer Research Foundation Dale F. Frey Award for Breakthrough Scientists 2302-17 (https://www.damonrunyon.org/), and startup funds from Carnegie Mellon University to AAB. WLN and AS were supported by NIH grant R01AI121337. LAC was supported by NIH grant R35GM159731 and startup funds provided by the University of Pittsburgh. MRP was supported by the CMLH Fellowship for Digital Health Innovation. The funders had no role in study design, data collection and analysis, decision to publish, or manuscript preparation.

## Author contributions

E.N. and A.A.B. were responsible for conceptualization. experimentation, data curation, investigation, methodology, validation, visualization, and writing the manuscript. I.V.M. was responsible for experimentation and data curation. M.R.P. and N.L.H. were responsible for methodology, formal analysis and visualization. A.S. and W.L.N. were responsible for mouse experimentation and analysis. C.A. and L.A.C. were responsible for experimentation, analysis and visualization. A.A.B. was responsible project administration and supervision. All authors contributed to writing, reviewing and editing the manuscript.

## Competing interests

The authors declare no competing interests.
