## [Transparent Peer Review file · Nature Communications]

A small periplasmic protein governs broad physiological adaptations in *Vibrio cholerae* via regulation of the DbfRS two-component system

Corresponding Author: Professor Andrew Bridges

Version 0:

Reviewer comments:

Reviewer #1

(Remarks to the Author)

The manuscript by Nguyen, et al. presents a convincing story about a two-component signaling pathway, DbfRS, which responds to envelope stress. The authors discovered a small protein, DbfQ, that greatly impacts DbfRS signaling and is directly involved in the signaling pathway. The authors also characterized phenotypes resulting from altered Dbf activity, such as changes to biofilm formation and motility. The involvement of c-di-GMP was also investigated. The techniques used are convincing and the manuscript is generally well written. However, there are some major concerns about the novelty and the incompleteness of the signaling pathway characterization.

-Major concerns-

One major concern is that the manuscript presents two loosely connected and mechanistically incomplete narratives: (1) DbfQ regulates DbfS activity, and (2) broader Dbf-regulated phenotypes. Both sections remain largely descriptive. The modulation of two-component systems by periplasmic proteins is well established—for example, ChvE's effect on the VirA/VirG system in *Agrobacterium* (see the 2010 review, PMID: 19943903). While the authors demonstrate an interaction between DbfQ and DbfS, they do not provide evidence showing how this interaction influences DbfS activity. In the second part, no direct targets of DbfR have been identified beyond its autoregulatory binding to its own promoter. As a result, the manuscript offers only incremental advances compared to the authors' previous excellent PNAS publication.

The authors do not clearly demonstrate the physiological significance of DbfQ regulation of DbfSR activity. It remains unclear why *V. cholerae* would require an additional regulator to modulate DbfSR and downstream processes such as biofilm dispersal. Furthermore, while the authors provide comparative genomic alignments showing dbfSQR-like arrangements in other bacteria, it would strengthen the manuscript to experimentally validate this conservation in at least one additional system, even through an *in vitro* approach.

Line 159. This assay does not definitively prove periplasmic localization. Signal along the outside of the cell could instead indicate that DbfQ is found within the inner membrane, or attached to the inner membrane on the cytoplasmic side. This article (PMID: 28667608) describes a method for more definitive demonstration of periplasmic localization.

While it is understandable that the authors chose to test various phenotypes of the DbfRD51V mutant in combination with dbfQ or dbfS, given the availability of DbfRD51V, this approach adds limited value. Since DbfRD51V is a loss-of-function mutant, the double mutant analyses do not provide additional mechanistic insight. Instead, the authors should consider generating a constitutively phosphorylated variant of DbfR—or preferably DbfS—to more directly assess whether DbfQ influences the phosphorylation relay.

A missing piece of this study is the nature of the signal detected by DbfQ. This is briefly mentioned in line 259. Identifying this signal would greatly increase the completeness of the manuscript.

Lines 463-465. The possibility of therapeutics targeting DbfQRS is of questionable novelty, as this sensory pathway responds to damage to membrane integrity, and a clinically available antibacterial targeting membrane integrity, polymyxin B, is already available and was mentioned in this paper. This antibiotic was found to induce DbfQRS greatly, meaning an intervention targeting this pathway is already available. The authors need to expand on this further to indicate how new therapeutics would be distinct from existing antimicrobials.

Lines 177-189. This portion of the story would be more complete if specific residues mediating binding were identified. If the AlphaFold simulation identified specific interacting residues between DbfS and DbfQ, these could be mutated to see if the interaction is abolished.

- Minor concerns -

Fig 1 legend. Parenthesis missing before “F”

Please give statistical significance values for Fig 1G, even though it appears obvious.

Line 84. This phrasing leads the reader to think a “Pfam domain” is a class of protein domain. Please rephrase to clarify that Pfam is a database, and that PepSY is the domain.

Figure 2B. Please give statistical significance values.

Lines 204-205. Please rephrase to clarify that the individual protein structures and their interaction are both AlphaFold simulations. Suggestion: “Predicted interaction between AlphaFold3 structures of DbfQ and full length...”

Fig 3. Please give statistical significance values.

Fig 4D. Please clarify that the asterisks refer to P-values compared to the wild type data set.

Lines 320-324. This explanation of the processes involved in regulating c-di-GMP levels may fit better as a paragraph in the introduction section.

Lines 328-332. Are the authors certain that the only factor that can cause the rugose colony phenotype is increased biofilm formation? Other factors can alter colony morphology, and most of these transposon mutants do not appear to have been investigated further with more biofilm-specific assays.

Line 425-431. This section is, in my opinion, too speculative. The signals detected by direct binding to DbfQ/S are not investigated in this study.

Line 433. Peptidoglycan biosynthesis is mentioned here and a couple more times in the text, however, this is not investigated in much detail in the study, apart from identifying genes through RNAseq. In the absence of additional experiments, the authors should not claim to have strong evidence of the roles of DbfQRS on peptidoglycan.

Line 487. “and” should be moved to before ampicillin.

There is inconsistency in the formatting of journal abbreviations in the references section.

Reviewer #2

(Remarks to the Author)

The study by Nguyen et al. characterizes the regulation and functional activities of the DfbRS two-component system in *Vibrio cholerae*. In particular, they show that DfbRS activity is regulated by a co-transcribed periplasmically localized PepSY-domain containing protein. Through a genetic screen, they uncover that DfbQRS signaling is stimulated by outer membrane perturbation. The authors also explore the regulatory consequences of DfbQRS signaling, which revealed that this system likely influences flagellar motility and cyclic-di-GMP signaling. Through another genetic screen, they identified that one diguanylate cyclase, CdgL, is necessary for the disruption of biofilm dispersal exhibited by mutants with overactive DfbRS signaling. Finally, they show that overactive DfbQRS signaling attenuates *Vibrio cholerae* during infection. Altogether, this manuscript is timely, thorough, well-written, clearly presented, and represents an important advance to the field. However, I have a few points for the authors to consider to enhance the clarity of the manuscript and to solidify their findings:

For the genetic modules identified in other species, are the PepSY domains predicted to interact with the nearby histidine kinase by AlphaFold3? If so, this would strengthen the claim that these are bona fide examples of regulatory modules outside of the Vibrionaceae.

While I appreciate that searching for the signal that induces DfbQRS activity should be the focus of future work, I had one suggestion for the authors to consider. One possibility is that the signal is an intermediate generated during membrane stress, as proposed. However, isn't it also possible that the signal is simply a media component that is only able to access the periplasm when the outer membrane is compromised? This could be tested by altering the media composition to see what effect it has on DfbQRS activity (e.g., remove casamino acids, glucose, etc.). Or at the very least, this should be acknowledged as a possibility in the text.

Fig. S6 and Fig. 4D – It is difficult to interpret whether these results actually reflect differences in motility. These types of motility assays inherently rely on bacterial growth and chemotaxis, and the authors have already demonstrated that Δ dfbQ and Δ dfbS strains exhibit reduced growth in Fig. S1. So, it would be more prudent to test motility using an assay like microscopy to assess the motility of single cells, which would not be confounded by differences in growth. If the transcription of flagellar assembly is decreased in these backgrounds as suggested by RNA-seq, then the frequency of motile cells should be decreased in the Δ dfbQ and Δ dfbS mutants.

Fig. 5 – these results suggest that cyclic-di-GMP influences one of the output activities associated with DfbQRS signaling – biofilm dispersal. But it is unclear whether cyclic-di-GMP is acting upstream of DfbQRS signaling, downstream of DfbQRS signaling due to CdgL diguanylate cyclase activity as proposed, or perhaps a combination of both. To test this, the authors could assess P_{dfbQRS}-lux reporter activity in Δ dfbS Δ cdgL vs Δ dfbS. If their hypothesis is correct, then reporter activity should be similar in both backgrounds.

Reviewer #3

(Remarks to the Author)

Version 1:

Reviewer comments:

Reviewer #1

(Remarks to the Author)

The revised paper from the Bridges group has addressed the majority of my concerns with the original version. The manuscript is more convincing and complete. Additional experiments such as the PhoA reporter for periplasmic localization supported the authors' previous results. Furthermore, although identification of the signal would be ideal, I believe it is acceptable for this to be the topic of a future manuscript, given the inherent difficulty in identifying this type of signal. Additionally, sections of the manuscript that were originally confusing have been clarified. My remaining critiques are of minor importance.

Line 230 and Fig S3A. Expression levels or protein abundance of the various mutants appears to be drastically different, despite all mutants being expected to disrupt salt bridges. This should be discussed.

Line 260. *P. aeruginosa* should be italicized

Line 365. It would be informative to give the results as proportion of cells that are motile for each condition (in %) rather than only saying that a "high frequency" of cells were not motile.

Line 856. Please give the diameter of the glass beads used.

Reviewer #2

(Remarks to the Author)

The authors have expertly addressed my comments and concerns from the first round of review. I believe that this manuscript is ready for publication at Nature Communications.

Reviewer #3

(Remarks to the Author)

Below are our point-by-point responses to the reviewers' comments. Reviewer comments are in black text and our responses are in blue text. We hope the manuscript is now acceptable for publication in *Nature Communications*. Please let us know if you need anything more.

Yours,

Drew Bridges

REVIEWER COMMENTS

Reviewer #1 (Remarks to the Author):

The manuscript by Nguyen, et al. presents a convincing story about a two-component signaling pathway, DbfRS, which responds to envelope stress. The authors discovered a small protein, DbfQ, that greatly impacts DbfRS signaling and is directly involved in the signaling pathway. The authors also characterized phenotypes resulting from altered Dbf activity, such as changes to biofilm formation and motility. The involvement of c-di-GMP was also investigated. The techniques used are convincing and the manuscript is generally well written. However, there are some major concerns about the novelty and the incompleteness of the signaling pathway characterization.

We thank the reviewer for thoroughly reading our manuscript, recognizing the contributions of our study, and providing constructive feedback.

-Major concerns-

One major concern is that the manuscript presents two loosely connected and mechanistically incomplete narratives: (1) DbfQ regulates DbfS activity, and (2) broader Dbf-regulated phenotypes. Both sections remain largely descriptive. The modulation of two-component systems by periplasmic proteins is well established—for example, ChvE's effect on the VirA/VirG system in *Agrobacterium* (see the 2010 review, PMID: 19943903). While the authors demonstrate an interaction between DbfQ and DbfS, they do not provide evidence showing how this interaction influences DbfS activity. In the second part, no direct targets of DbfR have been identified beyond its autoregulatory binding to its own promoter. As a result, the manuscript offers only incremental advances compared to the authors' previous excellent PNAS publication.

We appreciate the reviewer's concern and would like to clarify the novelty and mechanistic depth of this study compared to our prior work. Our earlier PNAS paper

established the link between DbfRS and biofilm dispersal but did not identify upstream inputs, modulators, the regulon, or consequences in the infection model. Indeed, in that study, only a single figure focused on DbfRS.

In the present manuscript, we delineate the signaling pathway from input to output—showing how DbfQ influences DbfS activity and, in turn, how DbfR drives transcriptional changes that control bacterial lifestyle transitions. Specifically, we characterize a small periplasmic protein, DbfQ, that had never been studied before. Using extensive genetic, biochemical, and biophysical approaches, we demonstrate that DbfQ directly interacts with the DbfS sensor kinase promoting its phosphatase activity. We present evidence that DbfQ-like proteins represent a new class of TCS regulators, often co-encoded with signaling machinery. While modulation of TCSs by periplasmic proteins has precedent, our findings extend this paradigm by defining a distinct mechanism and evolutionary context. Beyond autoregulation, we map the DbfR regulon (>500 genes) under controlled phosphorylation states, revealing coordinated regulation of biofilm formation, motility, and metabolism. Rather than focusing on direct DbfR binding sites, we chose to use targeted functional assays (motility, c-di-GMP levels, identification of the diguanylate cyclase CdgL, and infection fitness) to connect the regulon to phenotypic outcomes.

Taken together, our findings present a coherent systems-level signaling model that links DbfQ-mediated modulation of DbfS with DbfR-dependent transcriptional programs and their physiological consequences. We believe this integrated perspective represents a substantive conceptual and mechanistic advance over our prior work and contributes broadly to the understanding of bacterial signaling. In response to the reviewer's request, we have revised the manuscript to emphasize these mechanistic connections, provide additional experiments, and clarify our discussion (see below).

The authors do not clearly demonstrate the physiological significance of DbfQ regulation of DbfSR activity. It remains unclear why *V. cholerae* would require an additional regulator to modulate DbfSR and downstream processes such as biofilm dispersal. Furthermore, while the authors provide comparative genomic alignments showing dbfSQR-like arrangements in other bacteria, it would strengthen the manuscript to experimentally validate this conservation in at least one additional system, even through an in vitro approach.

We appreciate the reviewer's suggestion that validating the conservation of DbfQRS system across species would strengthen our study. In response to this point, and in line with Reviewer #2's comment, we have performed AlphaFold-based prediction of DbfQ- and DbfS-like proteins from *P. aeruginosa* to assess the potential for conserved interactions (Fig. S2A in the revised manuscript). We also attempted to purify the *P. aeruginosa* orthologs, but were unsuccessful after many attempts due to low protein solubilities. We have added the following information to the text:

Line 251: Given the cross-taxa conservation of DbfQRS system as shown by our bioinformatic analysis (Fig. 1E, Fig. S2A), we wondered whether orthologs of DbfS and DbfQ are also predicted to interact by AlphaFold3. We focused on the system encoded in P. aeruginosa. Interestingly, P. aeruginosa possesses two DbfQ-like proteins that are encoded adjacent to a two-component system. The PepSY protein PA2658, encoded immediately upstream of the sensor BqsS, was predicted to interact with BqsS with high confidence (ipTM = 0.89, notably higher than the score for the DbfQ-DbfS prediction). In contrast, the other PepSY protein, PA2659, encoded two genes upstream, was not predicted to bind with BqsS (ipTM = 0.11), potentially suggesting a selective interaction between PA2658 and the BqsS sensory domain in P. aeruginosa (Fig. S2B). Future studies will be required to determine whether PA2658 and PA2659 function in tandem or compete for the regulation of BqsS.

Why *V. cholerae* requires an additional regulator (DbfQ) to modulate DbfRS and downstream processes is an ongoing area of research for our group. As the referee points out below, we speculate that DbfQ binds a ligand that affects signal transduction through DbfS. This is discussed in the revised manuscript (Line 508).

Line 159. This assay does not definitively prove periplasmic localization. Signal along the outside of the cell could instead indicate that DbfQ is found within the inner membrane, or attached to the inner membrane on the cytoplasmic side. This article (PMID: 28667608) describes a method for more definitive demonstration of periplasmic localization.

We have verified periplasmic localization using the PhoA-lacZ α reporter as requested. Specifically, we found that the *dbfQ-phoA-lacZ α* strain exhibits blue pigmentation upon addition of X-Pho (5-bromo-4-chloro-3-indolyl-phosphate), indicating periplasmic localization (Fig. 2B of the revised manuscript). Positive and negative controls were included to validate our results. We have reported this result in our revised manuscript:

Line 183: To confirm the periplasmic localization of DbfQ, we fused dbfQ to a phoA-lacZ α reporter, which resulted in substantial periplasmic PhoA enzyme activity (Fig. 2B)³⁸.

While it is understandable that the authors chose to test various phenotypes of the DbfRD51V mutant in combination with *dbfQ* or *dbfS*, given the availability of DbfRD51V, this approach adds limited value. Since DbfRD51V is a loss-of-function mutant, the double mutant analyses do not provide additional mechanistic insight. Instead, the authors should consider generating a constitutively phosphorylated variant of DbfR—or preferably DbfS—to more directly assess whether DbfQ influences the phosphorylation relay.

While we agree that a constitutively phosphorylated allele of DbfS or DbfR could be useful in certain contexts, we stand by the logic for using the phospho-dead mutant in our experiments. In the $\Delta dbfQ$ strain, DbfR is already phosphorylated due to the kinase activity of DbfS, as shown by Phos-tag gel analysis (Fig. 1). Introducing a constitutively

phosphorylated DbfR into this background would not clarify DbfQ's role, as both single and double mutants would be expected to exhibit the same phenotypes driven by high DbfR-P levels. We used the *dbfR^{D51V}* allele, which cannot be phosphorylated, not simply because it was available, but because it allows us to test whether $\Delta dbfQ$ and $\Delta dbfS$ phenotypes depend on DbfR phosphorylation. Consistent with our model that DbfQ promotes the phosphatase activity of DbfS, *dbfR^{D51V}* completely suppresses the biofilm and reporter phenotypes of both $\Delta dbfQ$ and $\Delta dbfS$ strains, restoring outputs to wild-type levels (Fig 1D & G). These findings demonstrate that the observed phenotypes are mediated through DbfR phosphorylation, supporting our conclusion that DbfQ regulates DbfS-DbfR signal transduction. We have adjusted the text to better clarify this point:

*Line 89: Our next goal was to determine whether DbfQ controls the biofilm lifecycle through the DbfRS signaling cascade. To assess this possibility, we introduced the $\Delta dbfQ$ deletion into a genetic background carrying an allele of DbfR that is incapable of phosphorylation (the phospho-dead allele *dbfR^{D51V}*). We reasoned that if the extreme biofilm phenotype of $\Delta dbfQ$ is due to elevated phosphorylation of DbfR, then combining this mutant with the phospho-dead allele should abolish this phenotype. Indeed, we found that the $\Delta dbfQ$ *dbfR^{D51V}* double mutant lost the hyper-biofilm phenotype observed in $\Delta dbfQ$ single mutant (Fig. 1D), supporting a model in which DbfQ regulates biofilm dynamics through the DbfRS signaling pathway, potentially by modulating DbfS activity.*

A missing piece of this study is the nature of the signal detected by DbfQ. This is briefly mentioned in line 259. Identifying this signal would greatly increase the completeness of the manuscript.

We share the reviewer's interest in determining direct signal(s) sensed by DbfQ (and DbfS). Given the potential wide range of chemical or physical cues directly sensed by TCSs, definitive signal identification often requires extensive screens, chemical fractionation, or structural analyses that extend well beyond the scope of the present work (and could take many years). We therefore respectfully propose that signal identification represents a logical future direction.

Lines 463-465. The possibility of therapeutics targeting DbfQRS is of questionable novelty, as this sensory pathway responds to damage to membrane integrity, and a clinically available antibacterial targeting membrane integrity, polymyxin B, is already available and was mentioned in this paper. This antibiotic was found to induce DbfQRS greatly, meaning an intervention targeting this pathway is already available. The authors need to expand on this further to indicate how new therapeutics would be distinct from existing antimicrobials.

We have modified the discussion to clarify our logic. While polymyxin B partially activates DbfQRS, it does so via membrane damage and possesses broad antibacterial activity limiting its specificity and safety. In contrast, our study opens the possibility of identifying

small molecules that act directly on pathway components (DbfQ or DbfS) independent of membrane disruption (or killing cells). Such molecules could be used to drive DbfR phosphorylation, pushing bacteria into a maladaptive low-fitness state. This mechanistic specificity distinguishes our proposed therapeutic strategy from existing agents like polymyxin B, which have established DbfQRS-independent effects on bacterial physiology. The modified discussion reads:

Line 542: More broadly, understanding how stress-responsive TCSs like DbfQRS orchestrate bacterial adaptation may reveal novel antimicrobial strategies that exploit the inherent trade-offs in bacterial stress adaptation. Our findings present the DbfQ-DbfS interface as a potential site for the development of small-molecule modulators. Agents that directly disrupt DbfQ-DbfS binding and bias DbfS toward kinase output would hyperactivate the pathway, leading to a low-fitness state with reduced growth and colonization.

Lines 177-189. This portion of the story would be more complete if specific residues mediating binding were identified. If the AlphaFold simulation identified specific interacting residues between DbfS and DbfQ, these could be mutated to see if the interaction is abolished.

Based on our AlphaFold3 model, we have identified two potential electrostatic interaction within the DbfQ-DbfS interface, between DbfQ^{R84} – DbfS^{D62} and DbfQ^{K102} – DbfS^{E69}, and performed site-directed mutagenesis in these residues. We found that each of these point mutants exhibited a significant increase in $P_{dbfQRS-lux}$ reporter activity compared to WT, consistent with elevated DbfR phosphorylation due to weakened DbfQ-DbfS interaction (Fig. 3A&B of the revised manuscript). We have reported this experiment and its results in the revised manuscript:

Line 225: Closer inspection of the interaction interface revealed two putative electrostatic pairs, between DbfQ^{R84} – DbfS^{D62} and DbfQ^{K102} – DbfS^{E69} (Fig. 3A), that could mediate the interaction. We therefore introduced site-directed mutations in either dbfQ (dbfQ^{R84D} and dbfQ^{K102E}) or dbfS (dbfS^{D62R} and dbfS^{E69K}) to weaken the interaction and test their functional relevance. Western blot analysis confirmed robust expression of the mutant proteins (Fig. S3A), suggesting that mutagenesis did not destabilize either DbfQ or DbfS variants. Each point mutant yielded significantly increased $P_{dbfQRS-lux}$ reporter activity compared to the WT strain (Fig. 3B), suggestive of increased DbfR phosphorylation due to a weakened direct interaction between DbfQ and DbfS.

- Minor concerns -

Fig 1 legend. Parenthesis missing before “F)”

This is now fixed.

Please give statistical significance values for Fig 1G, even though it appears obvious.

This has been added.

Line 84. This phrasing leads the reader to think a “Pfam domain” is a class of protein domain. Please rephrase to clarify that Pfam is a database, and that PepSY is the domain.

We agree and have changed the wording accordingly to clarify this point:

Line 80: The mature ~12 kDa protein is predicted to contain a single domain, identified as 'PepSY' (residues 74–122) in the Pfam database³⁴.

Figure 2B. Please give statistical significance values.

This has been added. This is now Figure 2C.

Lines 204-205. Please rephrase to clarify that the individual protein structures and their interaction are both AlphaFold simulations. Suggestion: “Predicted interaction between AlphaFold3 structures of DbfQ and full length...”

We have made the suggested changes to reflect the reviewer’s comment. Now Line 263.

Fig 3. Please give statistical significance values.

This has been added.

Fig 4D. Please clarify that the asterisks refer to P-values compared to the wild type data set.

This has been corrected. This is now Figure 5D.

Lines 320-324. This explanation of the processes involved in regulating c-di-GMP levels may fit better as a paragraph in the introduction section.

Attempts to move this information to the introduction section disrupted the logical flow of the manuscript, so we prefer to keep a brief c-di-GMP introduction surrounding the results for CdgL.

Lines 328-332. Are the authors certain that the only factor that can cause the rugose colony phenotype is increased biofilm formation? Other factors can alter colony morphology, and most of these transposon mutants do not appear to have been investigated further with more biofilm-specific assays.

In *V. cholerae*, rugosity is consistently used as a metric for *vibrio* polysaccharide expression and is tightly linked to overproduction of biofilm matrix components (Fong *et al.*, 2010; Yan *et al.*, 2019). Indeed, the rugose phenotype was used to originally identify the biofilm matrix components in *V. cholerae* (see many works by Fitnat Yildiz group). We are not aware of other factors, outside of biofilm components, that impact rugosity in *V.*

cholerae. Accordingly, we used rugosity as a phenotypic indicator of hyper-biofilm in our screen and corroborated rugosity with biofilm-specific assays for the key follow-up strain ($\Delta cdgL$). The remaining transposon hits are presented as candidates for future study. We have adjusted the text to clarify this point:

Line 409: *This strain exhibits a wrinkled, “rugose” colony morphology, a vps-dependent phenotype linked to elevated biofilm matrix production in V. cholerae^{66,67}, which we use here as a practical phenotypic indicator of hyper-biofilm formation (Fig. 6C).*

Line 425-431. This section is, in my opinion, too speculative. The signals detected by direct binding to DbfQ/S are not investigated in this study.

Our intention for this discussion section is to outline two possible hypotheses directly motivated by our data. To avoid overspeculation, we have adjusted the text to state explicitly that the signals for DbfQRS remain unknown and label our ideas as possible working models for future investigation. This also accommodates Reviewer #2’s request to acknowledge that the signal could be an exogenous factor that accesses the periplasm only when the outer membrane is compromised.

Line 508: *Although the activating cue(s) for DbfQRS remain unknown, we suggest two possibilities that could explain its response to membrane damage. (1) DbfQ or DbfS could directly sense a cell-intrinsic feature of membrane damage, or (2) increased outer-membrane permeability could allow regulatory signal(s) to enter or escape from the periplasm. In either case, these signals could modulate the affinity of the DbfQ-DbfS interaction, and in turn DbfR phosphorylation. Future investigations will be required to resolve these hypotheses.*

Line 433. Peptidoglycan biosynthesis is mentioned here and a couple more times in the text, however, this is not investigated in much detail in the study, apart from identifying genes through RNAseq. In the absence of additional experiments, the authors should not claim to have strong evidence of the roles of DbfQRS on peptidoglycan.

As requested, we have removed all text discussing the connection between DbfQRS and peptidoglycan biosynthesis from the manuscript.

Line 487. “and” should be moved to before ampicillin.

This has been corrected.

There is inconsistency in the formatting of journal abbreviations in the references section.

We have made changes to the formatting of the References to ensure consistency.

Reviewer #2 (Remarks to the Author):

The study by Nguyen et al. characterizes the regulation and functional activities of the DfbRS two-component system in *Vibrio cholerae*. In particular, they show that DfbRS activity is regulated by a co-transcribed periplasmically localized PepSY-domain containing protein. Through a genetic screen, they uncover that DfbQRS signaling is stimulated by outer membrane perturbation. The authors also explore the regulatory consequences of DfbQRS signaling, which revealed that this system likely influences flagellar motility and cyclic-di-GMP signaling. Through another genetic screen, they identified that one diguanylate cyclase, CdgL, is necessary for the disruption of biofilm dispersal exhibited by mutants with overactive DfbRS signaling. Finally, they show that overactive DfbQRS signaling attenuates *Vibrio cholerae* during infection. Altogether, this manuscript is timely, thorough, well-written, clearly presented, and represents an important advance to the field. However, I have a few points for the authors to consider to enhance the clarity of the manuscript and to solidify their findings:

We'd like to thank the reviewer for recognizing the importance of our study and for providing their constructive feedback. In the revised manuscript, we have carried out additional experiments as suggested by the reviewer to strengthen our findings (see below).

For the genetic modules identified in other species, are the PepSY domains predicted to interact with the nearby histidine kinase by AlphaFold3? If so, this would strengthen the claim that these are bona fide examples of regulatory modules outside of the Vibrionaceae.

We thank the reviewer for this thoughtful suggestion. In line with Reviewer #1's comment, we have performed AlphaFold3-based predictions of DbfQ- and DbfS-like proteins in *P. aeruginosa* to assess the potential for conserved interactions. In *P. aeruginosa*, two PepSY proteins (PA2658, PA2659) lie upstream of the DbfS-like histidine kinase BqsS (PA2656). Our AlphaFold3 prediction shows a high-confidence PA2658–BqsS complex and a low-confidence PA2659–BqsS complex (Fig. S2B of the revised manuscript), consistent with selective PepSY–histidine kinase pairing and supporting our supposition that bona fide DbfQRS-like modules exist outside Vibrionaceae. We have added the following texts to reflect this result:

Line 251: Given the cross-taxa conservation of DbfQRS system as shown by our bioinformatic analysis (Fig. 1E, Fig. S2A), we wondered whether orthologs of DbfS and DbfQ are also predicted to interact by AlphaFold3. We focused on the system encoded in P. aeruginosa. Interestingly, P. aeruginosa possesses two DbfQ-like proteins that are encoded adjacent to a two-component system. The PepSY protein PA2658, encoded immediately upstream of the sensor BqsS, was predicted to interact with BqsS with high confidence (ipTM = 0.89, notably higher than the score for the DbfQ-DbfS prediction). In

contrast, the other PepSY protein, PA2659, encoded two genes upstream, was not predicted to bind with BqsS (ipTM = 0.11), potentially suggesting a selective interaction between PA2658 and the BqsS sensory domain in P. aeruginosa (Fig. S2B). Future studies will be required to determine whether PA2658 and PA2659 function in tandem or compete for the regulation of BqsS.

While I appreciate that searching for the signal that induces DfbQRS activity should be the focus of future work, I had one suggestion for the authors to consider. One possibility is that the signal is an intermediate generated during membrane stress, as proposed. However, isn't it also possible that the signal is simply a media component that is only able to access the periplasm when the outer membrane is compromised? This could be tested by altering the media composition to see what effect it has on DfbQRS activity (e.g., remove casamino acids, glucose, etc.). Or at the very least, this should be acknowledged as a possibility in the text.

We thank the reviewer for raising this interesting point. We tested different media compositions (without casamino acids, without glucose, with low/high Mg²⁺, and with low/high Ca²⁺ concentration) in the WT and outer-membrane defective $\Delta wavA$ strains. However, we observed no major difference in P_{dbfQRS}-lux output in any of these conditions. We note that these data do not exclude that other exogenous, periplasm-accessible cue(s) could be detected by DfbQRS. We now acknowledge this possibility in the text:

Line 508: Although the activating cue(s) for DfbQRS remain unknown, we suggest two possibilities that could explain its response to membrane damage. (1) DfbQ or DfbS could directly sense a cell-intrinsic feature of membrane damage, or (2) increased outer-membrane permeability could allow regulatory signal(s) to enter or escape from the periplasm. In either case, these signals could modulate the affinity of the DfbQ-DfbS interaction, and in turn DfbR phosphorylation. Future investigations will be required to resolve these hypotheses.

Fig. S6 and Fig. 4D – It is difficult to interpret whether these results actually reflect differences in motility. These types of motility assays inherently rely on bacterial growth and chemotaxis, and the authors have already demonstrated that $\Delta dfbQ$ and $\Delta dfbS$ strains exhibit reduced growth in Fig. S1. So, it would be more prudent to test motility using an assay like microscopy to assess the motility of single cells, which would not be confounded by differences in growth. If the transcription of flagellar assembly is decreased in these backgrounds as suggested by RNA-seq, then the frequency of motile cells should be decreased in the $\Delta dfbQ$ and $\Delta dfbS$ mutants.

To address this concern, we have performed single-cell motility assay as suggested by the reviewer. As the reviewer predicted, a large population of cells in the $\Delta dfbS$ and $\Delta dfbQ$ backgrounds did not exhibit motility (Fig. S6B of the revised manuscript). The modified text regarding this experiment now reads as follows:

Line 364: Furthermore, single-cell imaging of motility in the $\Delta dbfS$ and $\Delta dbfQ$ mutant strains revealed that a high-frequency of cells did not exhibit motility (Fig. S6B).

Fig. 5 – these results suggest that cyclic-di-GMP influences one of the output activities associated with DfbQRS signaling – biofilm dispersal. But it is unclear whether cyclic-di-GMP is acting upstream of DfbQRS signaling, downstream of DfbQRS signaling due to CdgL diguanylate cyclase activity as proposed, or perhaps a combination of both. To test this, the authors could assess PdfbQRS-lux reporter activity in $\Delta dbfS \Delta cdgL$ vs $\Delta dbfS$. If their hypothesis is correct, then reporter activity should be similar in both backgrounds.

To this point, we further clarified the relationship between CdgL and DbfQRS signaling. In the revised manuscript, we have measured and compared the $P_{dbfQRS-lux}$ activity in the $\Delta dbfS \Delta cdgL$ double mutant and the $\Delta dbfS$ single mutant backgrounds. Our data reveal no significant difference in reporter output between the two strains, supporting that c-di-GMP acts downstream of DbfQRS signaling due to CdgL activity. (Fig. S7C of the revised manuscript). We have added this result to the revised text:

Line 435: Additionally, the $\Delta cdgL$ single mutant exhibited $P_{dbfQRS-lux}$ activity indistinguishable from WT, whereas the $\Delta dbfS \Delta cdgL$ double mutant exhibited a significant increase relative to WT and was comparable to the $\Delta dbfS$ single mutant (Fig. S7C). These results indicate that CdgL functions downstream of DbfQRS signaling.

Reviewer #3 (Remarks to the Author):

REVIEWERS' COMMENTS

Reviewer #1 (Remarks to the Author):

The revised paper from the Bridges group has addressed the majority of my concerns with the original version. The manuscript is more convincing and complete. Additional experiments such as the PhoA reporter for periplasmic localization supported the authors' previous results. Furthermore, although identification of the signal would be ideal, I believe it is acceptable for this to be the topic of a future manuscript, given the inherent difficulty in identifying this type of signal. Additionally, sections of the manuscript that were originally confusing have been clarified. My remaining critiques are of minor importance.

We thank the reviewer for thoroughly reading and providing constructive feedback for our revised manuscript.

Line 230 and Fig S3A. Expression levels or protein abundance of the various mutants appear to be drastically different, despite all mutants being expected to disrupt salt bridges. This should be discussed.

The increased levels of mutant proteins result from positive feedback within the signaling cascade. Specifically, activation of the pathway (by loss-of-function *dbfS* or *dbfQ* mutations) can enhance expression of its own components (i.e. *dbfQ* and *dbfS*), leading to elevated protein abundance. We have added the following text to clarify this point:

“We note that the observed increased levels of DbfQ and DbfS mutant proteins results from positive feedback regulation within the cascade, which amplifies loss-of-function mutant protein expression.”

Line 260. *P. aeruginosa* should be italicized

This is now fixed.

Line 365. It would be informative to give the results as proportion of cells that are motile for each condition (in %) rather than only saying that a “high frequency” of cells were not motile.

We agree and have added the following text in the legend of Figure S6 to clarify this point:

“For $\Delta dbfS$ and $\Delta dbfQ$ mutant cells, >60% of cells exhibited swimming velocities less than 20 $\mu\text{m}/\text{sec}$.”

Line 856. Please give the diameter of the glass beads used.

The diameter of the glass beads used in this experiment is 4 mm. We have updated this information in the Methods section.

Reviewer #2 (Remarks to the Author):

The authors have expertly addressed my comments and concerns from the first round of review. I believe that this manuscript is ready for publication at Nature Communications.

We thank the reviewer for positive evaluation of our revised manuscript.

Reviewer #3 (Remarks to the Author):

We appreciate the reviewer's time and effort in evaluating our manuscript.